# Pessimistic Verification for Open-Ended Math Questions

Yanxing Huang [1]  Zihan Tang [* 1]  Zejin Lin [* 1]  Peng Li [2]  Yang Liu [2 3]

## Abstract

Automatic verification is a critical component in building math-solving agents and reinforcement learning, yet it often falls short in generalizability, performance, and cost-efficiency. Identifying that the primary bottleneck of verification lies in error detection capability, we propose pessimistic verification, a paradigm of agentic workflows that rejects a solution if any of multiple parallel verifiers identifies a flaw. We further introduce progressive pessimistic verification, which employs fine-grained proof decomposition to significantly enhance verification accuracy and efficiency. Our approach surpasses the performance and token efficiency of extended long chain-of-thought (long CoT) and mainstream verification workflows, crucially, our analysis reveals that existing benchmarks underestimate its effectiveness on stronger models due to inherent annotation errors. To further validate the effectiveness of our method, we applied a verification-based solving workflow on the IMO 2025 and MathArena Apex 2025 datasets, where the workflow with progressive pessimistic verification exhibits remarkable improvements in both efficiency and accuracy on highly challenging contest-level math problems with state-of-the-art models. Code is available at `https://github.com/THUNLP-MT/pverify`.

## 1. Introduction

Since the release of OpenAI o1 (OpenAI et al., 2024) and DeepSeek-R1 (DeepSeek-AI et al., 2025a), LLMs have achieved remarkable success in mathematical reasoning tasks. Beyond this achievement, their capability to verify mathematical proofs is also becoming increasingly important. This is primarily reflected in the following aspects:

Verification lies in the central place of almost all agentic workflows for math problem solving (Liu et al., 2025; Shao et al., 2025; DeepSeek-AI et al., 2025b; Huang & Yang, 2025; Varambally et al., 2025; Chen et al., 2025b; Achim et al., 2025; Chen et al., 2025a). It is widely adopted and validated as an effective approach that enables further test-time scaling over long chain-of-thought. Some works claimed to achieve IMO gold-medal level performance (Shao et al., 2025; DeepSeek-AI et al., 2025b; Huang & Yang, 2025) where verification-based iterative workflows serve as the crucial enabler.

Automatic verification is equally critical in reinforcement learning for large reasoning models (DeepSeek-AI et al., 2025a; OpenAI et al., 2024). In many prominent recent studies, both rule-based (DeepSeek-AI et al., 2025a; OpenAI et al., 2024) and LLM-based automated verification (Shao et al., 2025; DeepSeek-AI et al., 2025b) methods have been employed during training, yielding remarkable effectiveness.

Despite its importance, automatic verification for mathematics remains fundamentally challenging:

First, rule-based verification methods are highly efficient and widely used in training, but they are restricted to problems whose answers can be deterministically checked, leaving most open-ended mathematical questions beyond their scope. They can not be used in agentic workflows for math either.

Second, the absolute accuracy of LLM-based automatic verification is not satisfying yet. While the advancement of LLMs has led to numerous AI systems assisting in mathematical research, these workflows still rely heavily on human verification and expert guidance (Liu et al., 2025; Bryan et al., 2026). A primary reason for this dependence is the inadequate self-verification capability of AI systems, which renders their outputs unreliable.

Third, computational costs of verification severely limit the applicability of AI systems in mathematical problem-solving. For instance, the workflows employed by DeepSeekMath-V2 (Shao et al., 2025) and DeepSeek-

*Equal contribution [1]Qiuzhen College, Tsinghua University, Beijing, China [2]Institute for AI Industry Research (AIR), Tsinghua University, Beijing, China [3]Dept. of Comp. Sci & Tech., Institute for AI, Tsinghua University, Beijing, China. Correspondence to: Peng Li <lipeng@air.tsinghua.edu.cn>, Yang Liu <liuyang2011@tsinghua.edu.cn>.

*Proceedings of the 43rd International Conference on Machine Learning*, Seoul, South Korea. PMLR 306, 2026. Copyright 2026 by the author(s).

V3.2 (DeepSeek-AI et al., 2025b) generate a vast number of candidate proofs for each problem, subjecting each to 64 rounds of parallel independent verification. Consequently, solving a single IMO-level problem can consume at least thousands of API calls. There is, therefore, an urgent need for more efficient methods for automatic verification.

We contend that the key limitation of automatic verification lies in the ability of finding errors in a proof, which is supported by a series of recent works (Pandit et al., 2025; Liu et al., 2025; Shao et al., 2025; Huang & Yang, 2025). This brings us to the concept of pessimistic verification. This method involves checking a proof multiple times via an LLM and marking it as incorrect the moment an error is found. On the other hand, intuition tells us that verification requires a careful examination of details rather than repeated rough checks. This leads to our new proof decomposition and piecewise verification approach.

Based on these ideas, we designed and examined three variants: simple, vertical, and progressive pessimistic verification. The simple variant is very similar to mainstream solutions for automatic verification, while the progressive variant consistently achieves the best performance and efficiency across all scenarios in our experiments. Its progressive decomposition strategy will also become imperative when dealing with lengthy, intricate proofs, which is commonly seen in research-level problems.

We primarily conducted our experiments on two verification benchmarks, *Hard2Verify* (Pandit et al., 2025) and *IMO-GradingBench* (Luong et al., 2025). They are both contest-level math verification benchmarks with expert annotations. We also conducted some experiments on a homemade undergraduate-level math problem dataset for verification performance, which effectively complements existing elementary math datasets.

Progressive pessimistic verification not only boosts performance across these datasets but also shows better scaling efficiency than long chain-of-thought. Notably, our case studies with state-of-the-art models uncovered many errors missed by the original labels, suggesting that these benchmarks are no longer reliable on cutting-edge LLMs.

Given the limitations of verification benchmarks, we further tested our method on IMO 2025 (Balunović et al., 2025) and MathArena Apex 2025 (Balunović et al., 2025) for agentic problem solving with self-verification. The results also confirm that progressive pessimistic verification brings significant gains in both performance and efficiency.

Our main contributions are:

- Our study highlights and investigates the cost inefficiencies widespread in current math reasoning workflows, providing a new perspective for the community.

- We proposed three variants of pessimistic verification methods that could significantly enhance the verification ability of large language models with a higher efficiency than scaling long CoT.

- We conducted comprehensive experiments to validate our approach and to determine the actual capabilities of frontier models on IMO-level tasks within a controlled budget.

## 2. Method

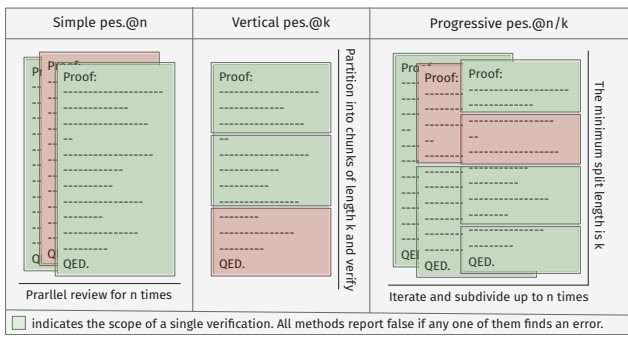

*Figure 1.* Three variants of pessimistic verification methods in this work. In following experiments they will separately be denoted as "pes@n", "vp@k", and "prog@n/k"

In Figure 1, we illustrate the three main pessimistic verification variants discussed in this paper. All three methods subject a single proof to multiple rounds of verification, adopting the first identified error as the final verification result. The primary distinction between them lies in the specific construction of these verification rounds. Here we will provide some detailed explanation for them.

### 2.1. Simple Pessimistic Verification

The most straightforward approach to automatic verification using an LLM is to directly prompt the model to inspect the entire proof for potential errors, which we refer to as standard verification. Simple pessimistic verification constructs its multiple rounds by simply repeating standard verification process for $n$ times in parallel.

This direct parallel verification scheme represents the current mainstream approach in verifier-aided problem solving and has been adopted by numerous related works (Liu et al., 2025; Shao et al., 2025; Huang & Yang, 2025). While the handling of verification results varies across these studies, they generally emphasize the importance of negative verdicts. However, previous works have not rigorously examined this approach as a standalone verification method. In this work, simple pessimistic verification can also be viewed as a representative baseline.

## 2.2. Vertical Pessimistic Verification

In spite of simply applying multiple reviews on the whole proof, we also explored a pessimistic verification from another dimension. As shown in Figure 2, we adopted a special prompting method and require the LLM to focus on a certain part of the proof, and try looking deep into these contents to find errors. Vertical pessimistic verification adopts a hyperparameter $k$, it first splits the whole proof into chunks with at most $k$ lines, and create a series of parallel review tasks for each chunk. Although this method only goes through the proof once, we also witnessed improved performance in error detection over single-pass whole proof verification and even simple pessimistic verification under similar computational budget.

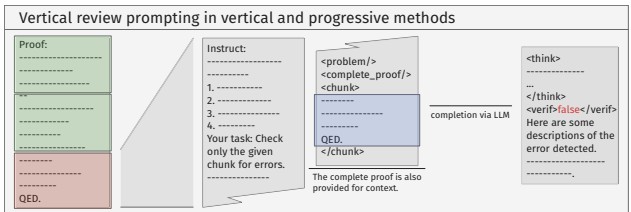

*Figure 2.* The vertical review prompting method used in vertical pessimistic verification and progressive pessimistic verification.

Concurrently, we also observed a positive scaling trend of vertical pessimistic verification as its segmentation becomes increasingly refined. However, hyperparameter selection remains a challenge. While fine-grained verification is necessary to detect subtle errors within proofs, applying it uniformly incurs a high cost for detecting easily discoverable errors. This trade-off inspired us to design a verification scheme capable of dynamically selecting segmentation granularity, leading to the development of progressive pessimistic verification.

## 2.3. Progressive Pessimistic Verification

Progressive pessimistic verification combines the mechanism of both simple and vertical methods. It performs multiscale verification, ranging from the level of the whole proof down to detailed steps similar to vertical pessimistic verification. This allows it to quickly filter out obvious errors in the proof, and then drill down into the detailed steps to carefully examine the correctness of each part.

Progressive pessimistic verification naturally supports pruning operations, thereby further improving its execution efficiency. It begins with iterative verification on a single complete proof, and after each iteration, the proof is subdivided in half and each part is evaluated separately in the manner shown in Figure 2. After every iteration, proofs in which errors have been detected are filtered out, and verification continues on the remaining proofs. We use the parameter $n$ to control the maximum number of iterations,

in which case the maximum number of checks performed on a single proof is $2^n - 1$. We also introduced a parameter $k$ to set a lower bound on the segmentation size, which prevents the proofs from being over-segmented in subsequent iterations.

This method combines the strengths of the other two and demonstrates the highest efficiency and performance ceiling in our experiments.

## 3. Experiments

### 3.1. Datasets and Metrics

In our experiments we primarily use three datasets for evaluation, and we constantly use the same prompt and workflow setting across all these dataset. Here are some descriptions about them:

- **IMO-GradingBench** (Luong et al., 2025). This dataset contains 1,000 human-graded solutions to IMO-level proof problems from IMO-ProofBench (Luong et al., 2025), from which we selected a subset with 300 samples for evaluation. This is a challenging test of fine-grained mathematical proof evaluation. Test results on the IMO-GradingBench subset mirror those of the complete dataset, thereby validating its representativeness for our experiments.

- **Hard2Verify** (Pandit et al., 2025). Hard2Verify contains 200 challenging problems and solutions from recent math competitions such as IMO and Putnam. The solutions are generated by strong models such as GPT-5 (Singh et al., 2025) and Gemini 2.5 Pro (Huang & Yang, 2025) and annotated by humans.

- **QiuZhen-Bench** (annotated by GPT-5). QiuZhen-Bench is a high-quality advanced math problem dataset. This dataset is sourced from challenging university-level math competitions and doctoral qualifying exams well-known for their difficulty. Given the limitation of human annotation discussed in Section 3.4, we used strong language models to generate proofs and annotations for this dataset. Then it is used to automatically evaluate weaker models. You can refer to Appendix A.2 for more details of the construction of this dataset.

All evaluation in our experiments was conducted at the response level, where we require the verifier to decide whether the whole proof is correct instead of providing a score or step-level evaluation. For IMO-GradingBench, only the responses that obtained fully 7 points are considered correct, otherwise they will all be considered false. And aside from our experiments, we will use standard verification and simple pessimistic verification as the baseline, as they are the

most representative among mainstream applications. In all our experiments we set the temperature parameter to 1.0 to ensure diversity of the responses.

In this work, we treat mathematical proof verification as a binary classification problem and focus on the following performance metrics:

- **True Negative Rate (TNR)**: The proportion of detected errors among all erroneous proofs. This is the primary metric for evaluating the model's ability to identify incorrect proofs.

- **True Positive Rate (TPR)**: The proportion of proofs classified as correct among all truly correct proofs. This helps assess the model's proof-verification capability.

- **Balanced F1 Score**: The harmonic mean of TPR and TNR, providing a balanced indicator of performance when both false positives and false negatives matter. This is also the primary indicator used in Pandit et al. (2025).

$$\text{Balanced F1} = \frac{2 \cdot \text{TPR} \cdot \text{TNR}}{\text{TPR} + \text{TNR}} \quad (1)$$

### 3.2. Preliminary Experiments

Prior to presenting the main experimental results, we have included results from preliminary experiments to justify our choice of experimental design.

Firstly, we briefly explore the mechanism behind the effectiveness of pessimistic mechanism. Since the intrinsic difficulty of proof verification lies in identifying errors, detecting an error during multi-round verification is inherently a low-probability event. Pessimistic verification methods effectively amplify the probability of this detection. Meanwhile, since current language models typically provide sound reasoning and explanations before pinpointing an error, actual false negatives are rare. Consequently, the overall verification performance is improved.

We can contrast this with majority voting, a widely popular test-time scaling method. It fails to help the model better identify errors, rendering it nearly ineffective for verification tasks (Pandit et al., 2025; Mahdavi et al., 2025b). Our preliminary experimental results, shown in Figure 3, corroborate this reasoning: Majority Voting is basically ineffective, whereas Pessimistic Verification methods demonstrate not only effectiveness but also stable scaling potential. And for this reason, we exclude majority voting from comparisons in our subsequent experiments.

We also benchmarked two representative LLMs against both the full and partial IMO-GradingBench sets. The consistent results shown in Table 1 confirm that the subset with 300

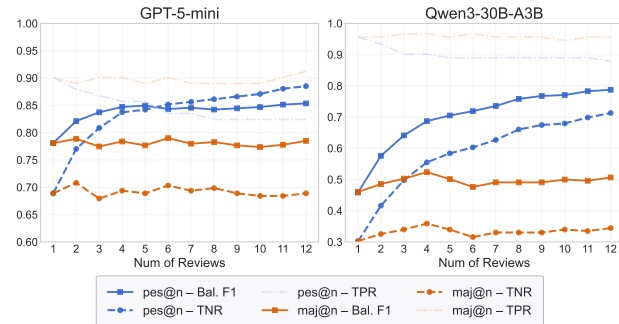

*Figure 3.* The comparison of simple pessimistic verification and majority voting on the subset of IMO-GradingBench. The former exhibits steady performance gains as sampling budget increases, whereas the latter shows almost no changes in performance.

questions, which is comparable in size to the other datasets, is sufficiently representative. Accordingly, all further experiments except for case studies on IMO-GradingBench were conducted using this subset.

*Table 1.* Comparison of the evaluation metrics of GPT-5-mini on the full set and subset of IMO-GradingBench. Results indicate that this subset of 300 questions is already representative for the verification performance. Here standard method refers to directly prompting the LLM to verify the given proof.

| Method | Size | TPR | TNR | Bal. F1 |
|---|---|---|---|---|
| standard | 1,000 | 0.917 | 0.660 | 0.768 |
| | 300 | 0.923 | 0.675 | 0.780 |
| progressive@3/6 | 1,000 | 0.850 | 0.816 | 0.833 |
| | 300 | 0.857 | 0.842 | 0.849 |

### 3.3. Main Results on Verification Benchmarks

We begin by conducting some scaling experiments to test the effect of hyperparameter settings in pessimistic verification methods. We primarily examined two representative models in these experiments, GPT-5-mini (Singh et al., 2025) and Qwen3-30B-A3B (Yang et al., 2025). They could reflect the performance gains of language models of medium and small sizes. We did not conduct scaling experiments on large models since these benchmarks are already unreliable for them.

The results of our scaling experiments are summarized in Figure 4. These experiments showed similar trend of our methods. All three types of pessimistic verifications constantly benefit from scaling up computational budget. The performance of the vertical pessimistic method appears to be more influenced by the specific dataset, but when the segmentation is sufficiently fine-grained, it does indeed improve performance while the proof has been read through only once. Its inefficiency stems largely from token wastage

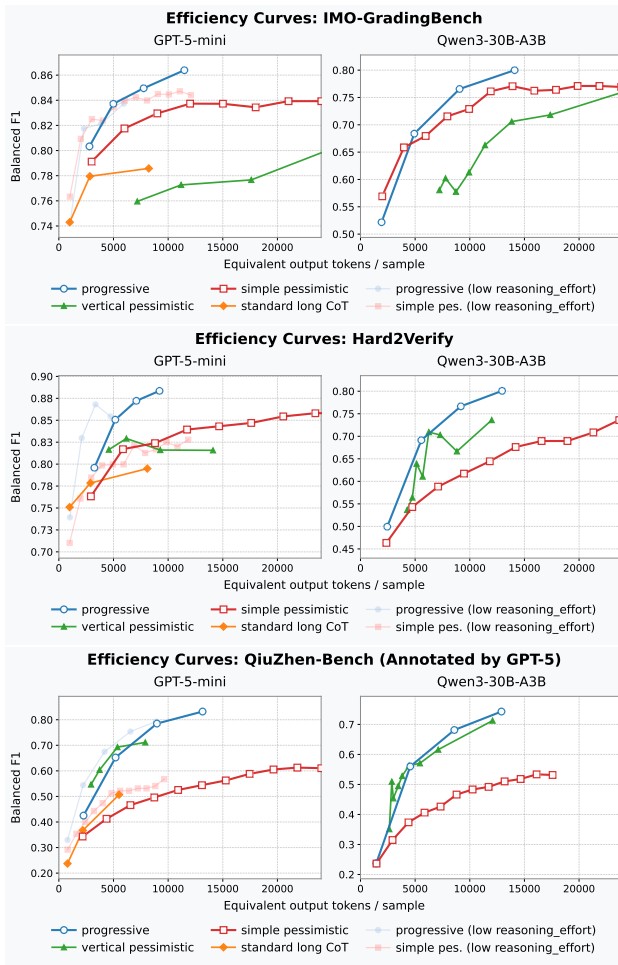

*Figure 4.* The response-level performance and efficiency of pessimistic verification methods on three verification datasets. The equivalent output tokens is calculated by weighting the model's input and output token prices. For GPT-5-mini and Qwen3-30B-A3B it is (Output + Input / 8). The reasoning effort of Qwen series models can not be adjusted so we excluded long-CoT results of them. These results indicate that pessimistic verification has stable and efficient scalability.

caused by fixed verification precision, a critical issue that can be effectively resolved through progressive segmentation. And as the combination of simple and vertical methods, progressive pessimistic verification constantly exhibits the highest performance and efficiency. Moreover, on models that the length of chain-of-thought is controllable (i.e. GPT-5-mini), simple and progressive methods constantly exhibits higher efficiency in test-time scaling than long CoT. Their parallel nature makes them even faster than long chains of reasoning.

Then we have also conducted vast experiments with standard and progressive pessimistic verification methods across different models on all three datasets, and the main results is summarized in Figure 5. In these experiments we use

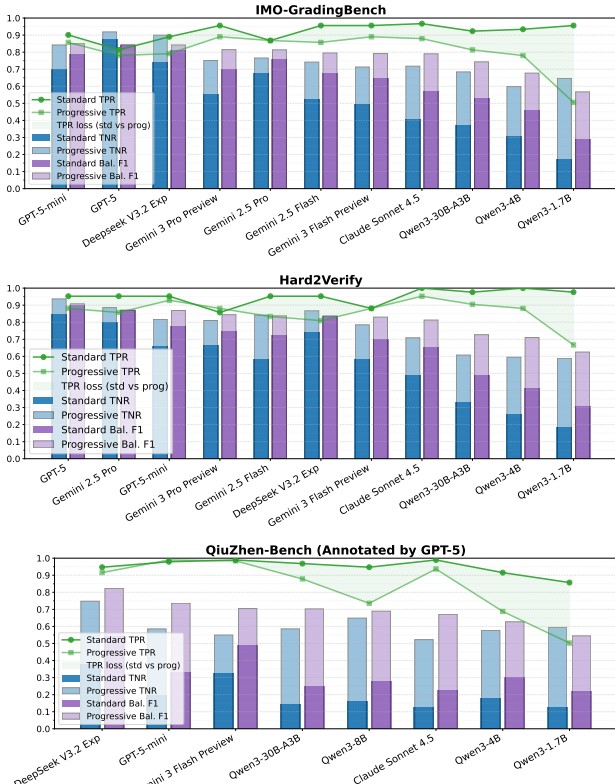

*Figure 5.* The main benchmark results on IMO-GradingBench, Hard2Verify and QiuZhen-Bench. Thinking mode is enabled for all models if possible, and the reasoning effort is set to medium when supported. This method constantly improves the classification performance across almost all tested models. Models are sorted by their Bal. F1 score under prog@3/6.

the hyperparameter setting progressive@3/6 for balanced performance and efficiency. From these graphs we can read several key findings:

- Error detecting is the key ability that separates strong verifiers from the weak which is also discovered in some recent works (Pandit et al., 2025). On all datasets the TPR is almost the same across all models from strong to weak, while TNR varies a lot.

- Merely from statistics we can see that progressive pessimistic verification can significantly increase TNR and balanced F1 score for most models, with a small tradeoff of TPR.

- Weaker models tend to exhibit higher performance gains with a greater degression of TPR. The consistency on positive samples is also an important indicator of the verification ability of LLMs.

- In fact, based on more fine-grained case studies in Section 3.4, we found that for stronger models, the

decrease in TPR is largely an illusion caused by dataset annotation, and the actual performance improvement is being underestimated.

### 3.4. Case Study

To better analyze performance beyond aggregated statistics and reflect more realistic behavior, we also conducted a detailed case study on the results of the pessimistic verification. In this evaluation, we focus primarily on the False Negative cases, because TP and TN generally represent correct classifications, while most FP cases occur simply because our verifier failed to detect existing errors in the proof. FN cases are usually the most informative.

The statistical results of our case study can be found in Table 2. In the Appendix A.4, we also provide several concrete examples of mislabeling in the original dataset, and the full set of cases we examined is available in our open-source code repository. Our classification criteria are based on whether the errors identified by these models in FN cases are:

- **Critical**. If our verifier does uncover a critical error that was overlooked in the original dataset annotation, or a difficult-to-resolve tricky skip, it directly affects the correctness of the entire proof.

- **Minor**. If our verifier identified a typo or minor errors that can be easily fixed. In such cases, whether it should be judged as an error depends entirely on subjective criteria.

- **Nonsense**. If our verifier misunderstands the problems and produces some unwarranted judgments. These are the truly harmful false negatives.

*Table 2.* Manual review of FN cases in progressive pessimistic verification on IMO-GradingBench. The classification is based on whether the judgments provided by the verifier in these FN cases are critical, minor, or nonsense. These cases are regarded as false in benchmarks, while a large percentage of them generated by state-of-the-art models actually discovered new critical errors ignored in the dataset annotation.

| Model | Critical | Minor | Nonsense | Total |
|---|---|---|---|---|
| Gemini 3 Pro Preview | 8 | 0 | 2 | 10 |
| GPT-5 | 9 | 9 | 2 | 20 |
| GPT-5-mini | 10 | 6 | 4 | 20 |
| Qwen3-30B-A3B | 1 | 1 | 18 | 20 |

We found that most of the FN cases produced by GPT-5-mini or stronger models under pessimistic verification did in fact identify real, critical errors in the proofs. These errors had been mistakenly labeled as correct by the dataset annotators. This indicates, on one hand, that our previous experiments substantially underestimated the verification capability of stronger models, and on the other hand, it supports the validity of using GPT-5 to annotate QiuZhen-Bench. Meanwhile, many smaller models do indeed produce a large number of unwarranted False Negative judgments.

### 3.5. Problem-Solving Experiments

Progressive pessimistic verification exhibits strong performance and efficiency in many proof evaluation benchmarks on language models of smaller sizes. However, all these benchmarks annotated by either human experts or LLMs fail to measure reliably the verification performance according to our case studies. To further promote the value of pessimistic verification methods, we have also done a series of experiments in problem solving with verification-based agentic workflows. We designed a simple iterative workflow similar to Huang & Yang (2025), which is shown in Figure 6. The effectiveness of pessimistic verification on them could be reflected by problem solving experiments here. We primarily used Gemini 3 series models for problem-solving experiments for their advanced reasoning ability.

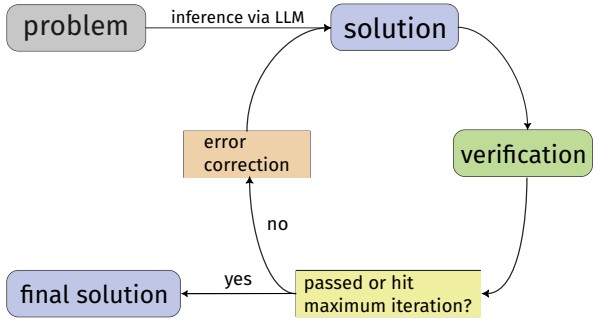

*Figure 6.* Flow diagram of our problem solving workflow. This pipeline is adopted by experiments on IMO 2025 and MathArena Apex 2025 dataset. Note that the number of iterations of this workflow is largely dependent on the performance of the verification method and the inherent problem-solving ability of the LLM.

We used two extremely hard datasets to evaluate the performance of problem solving with this workflow:

- **IMO 2025** (Balunović et al., 2025). This dataset contains 6 problems from the 66th International Mathematical Olympiad (IMO) contest at the year 2025. We collected the output proofs of our workflow and manually reviewed each output with the provided grading scheme.

- **MathArena Apex 2025** (Balunović et al., 2025). Apex 2025 contains 12 problems from various math competitions in 2025. These questions have been specially selected to exceed the average difficulty of IMO compe-

tition problems. They also come with standard answers, allowing for convenient automatic grading.

In our experiments on the IMO 2025 dataset, we selected the results from the final iteration of each method and performed manual evaluation on every problem. Conversely, for the Apex 2025 dataset, we plotted accuracy and cost curves across workflow iterations by verifying against standard answers. Given the small size of the Apex 2025 dataset and the potential for significant experimental variance, we repeated the tests three times for each setting and plotted all resulting curves within a single figure.

Furthermore, regarding token consumption, we calculate the cumulative tokens from every API call made until a proof is finally accepted, including the final successful verification step. This metric best reflects real-world usage scenarios. The reported cost is expressed as equivalent output tokens, a weighted figure based on API pricing. For the Gemini 3 model series, this is calculated as: Output + Input / 6.

*Table 3.* Performance comparison on the IMO 2025 dataset across different methods with Gemini 3 Pro Preview. Scores for Problems 1–6 (P1–P6) are reported out of 7. The token usage is measured by equivalent output tokens. Among three methods, "direct" refers to directly prompting the LLM to solve these problems without verification-based workflow.

| Method | direct | w/ pes@12 | w/ prog@4/4 |
|---|---|---|---|
| P1 | 2 | 2 | 6 |
| P2 | 0 | 0 | 0 |
| P3 | 0 | 6 | 7 |
| P4 | 5 | 5 | 5 |
| P5 | 6 | 4 | 4 |
| P6 | 0 | 0 | 0 |
| **Total Score** | **13** | **17** | **22** |
| **Token Usage** | 129K | 1,310K | 985K |

The performance of Gemini 3 pro with different verification methods is shown in Table 3, and the performance-efficiency curve of Gemini series models on Apex 2025 dataset is shown in Figure 7. We can read several observations by looking into the cases of these results:

Initially, while the Gemini 3 Pro model could arrive at the correct final solution on the IMO 2025 dataset, its derivation process is often marred by errors and gaps, precluding it from securing full marks during our evaluation. Subsequent self-verification and iterative refining yielded noticeable improvements, while progressive pessimistic verification exhibited the best performance and efficiency. Notably, Problem 6 (P6) in IMO 2025 remains particularly challenging for language models, likely due to its demand for substantial geometric intuition. To date, it remains unsolved across nearly all existing studies (Shao et al., 2025; Luong et al.,

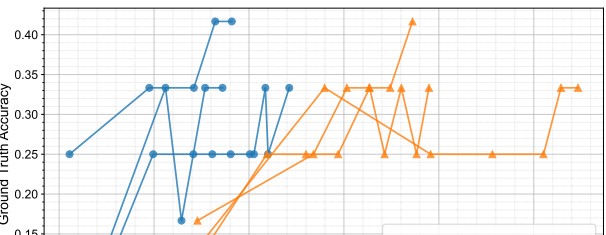
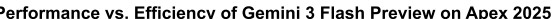
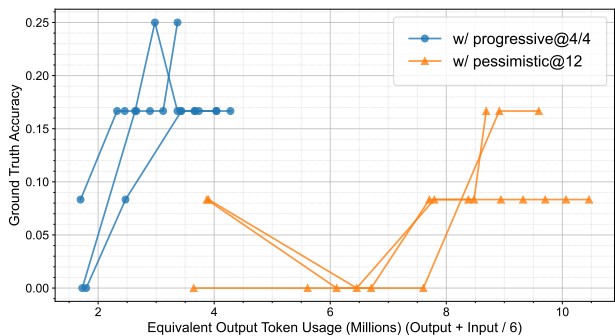

*Figure 7.* Performance of Gemini 3 series models on the Math-Arena Apex 2025 dataset using the iterative tuning workflow with different verification schemes. Given the limited dataset size and high testing costs, three experimental trials were conducted for each setting and plotted within the same figure. It is evident that performance improves significantly as iterations progress across all settings, with the progressive verification demonstrating notably higher efficiency.

2025; Achim et al., 2025).

At the same time, the performance gains on the Apex 2025 dataset are much more significant, owing to its higher average difficulty compared to IMO 2025. Although we observed instances where correct solutions were erroneously modified during the process, the final accuracy at the termination of the last iteration showed a significant improvement. Furthermore, our proposed progressive pessimistic verification scheme demonstrates clear efficiency advantages over the currently prevalent naive parallel verification methods.

# 4. Related Work

## 4.1. Informal Verification in Math

Using LLM for evaluation or verification tasks is a natural idea. At the early development stage of LLM, some works have already tried this method on traditional tasks in natural language processing (Liu et al., 2023; Zheng et al., 2023). The open-ended math problem lies between the math answering problem and other evaluation problems. It lacks

simple and direct means of verification, but its correctness is entirely objective. There are numerous works that focus on leveraging the internal capability of the LLM to achieve automatic evolution in mathematics (Liu et al., 2025; Huang & Yang, 2025; Xu et al., 2026; Shao et al., 2025; DeepSeek-AI et al., 2025b; Zuo et al., 2025). These methods yield significant performance improvements without depending on verifiable reward in math or code.

Some works in this area primarily focus on the alignment of LLM grading and that of humans. Certain agentic workflows could enhance this ability (Mahdavi et al., 2025a;b; Luong et al., 2025), while reinforcement learning for grading also exhibits effectiveness on small models (Dekoninck et al., 2025). However, this research direction differs from ours. While aligning grading criteria is a meaningful endeavor, it involves significant subjective factors, which diverges from the objective error detection we focus on. Some works also highlights the importance of error detection, as this is the key ability that separates strong verifiers from weaker ones (Pandit et al., 2025; Jain et al., 2025). But they did not build or validate a effective verification approach in math with this intuition. Before the release of this work there is no well-known and scalable method that could enhance this ability other than long CoT or multiple simple verifications with following process (Pandit et al., 2025).

### 4.2. Formal Verification

Another possible solution to verifying mathematical proofs is through formal theorem provers such as Lean (Chen et al., 2025b; Varambally et al., 2025; Chen et al., 2025a), which could provide completely reliable verification on math proofs. It is the verification method that distinguishes formal mathematics from informal mathematics.

Some works also claim the ability of achieving IMO gold-medal with formal provers and verification-based workflows (Hubert et al., 2025; Achim et al., 2025), while the overall performance of formal theorem provers still falls behind provers in natural language by a large margin (Dekoninck et al., 2025). Verification also dominates the cost of formal theorem proving. In formal proving, we are required to perform highly complex autoformalization of a proof to accommodate formal verification (Varambally et al., 2025; Chen et al., 2025b; Hubert et al., 2025; Achim et al., 2025). For example in Hilbert (Varambally et al., 2025) the workflow requires hundreds of API calls and millions of tokens on average only to solve one problem from Putnam dataset. While the informal verification discussed in this paper does not ensure total accuracy, it offers a viable compromise by trading strict rigor for greater ease of use and cost-efficiency. Given that the evolution of human mathematics has not invariably depended on absolute rigor, we posit that once informal verification achieves sufficient performance, it will

satisfy the requirements of most practical applications.

## 5. Conclusion and Discussion

In this work we proposed and analyzed pessimistic verification, among which progressive pessimistic verification exhibits strong performance and token efficiency on verification tasks. Its effectiveness is further validated in agentic problem-solving tasks with frontier LLMs.

Beyond the specific implementation of this mathematical proof verification scheme and the experimental results, we believe that two key underlying philosophies deserve greater attention:

- The first is the **error-centric verification** philosophy. Among verification outcomes, those identifying errors possess significantly higher information content, warranting greater attention and prioritization. Moreover, a strict pessimistic verification approach facilitates pruning, thereby further enhancing algorithmic efficiency.

- The second is the strategy of **segment-wise partitioning** and meticulous proof scrutiny. Whether verifying mathematical proofs or debugging code, it is essential to devote substantial effort to examining the minute details within the process, rather than merely conducting repeated, superficial reviews. This may be one of the reasons why the efficiency of progressive pessimistic verification significantly outperforms that of simple pessimistic verification.

Furthermore, the significance of our method and automatic verification may also be reflected in the following aspects:

- This method can further push the capability frontier of state-of-the-art large models, so there is an opportunity to incorporate it into the training pipeline to further raise the upper limit of large models' abilities in verification and rigorous reasoning tasks. This would be especially important for open-ended, long-form tasks (Sinha et al., 2025).

- In many software engineering applications, pessimistic verification may also be useful. We might also be able to use progressive verification to quickly identify potential issues in code without the need to design test cases or execute the code. Progressive segmentation will be a necessary technique since the context of code could be extremely lengthy.

- Aside from concrete use cases, the ability to verify should be viewed as a core part of general intelligence. Exploring methods to raise the ceiling of automatic verification and improve the fundamental abilities of LLMs remains an endeavor of enduring value.

## Acknowledgement

This work is supported by the National Natural Science Foundation of China (No. 62276152, 62236011), funding from Wuxi Research Institute of Applied Technologies, Tsinghua University under Grant 20242001120, and Key Laboratory of Ethnic Language Intelligent Analysis and Security Governance of MOE, Minzu University of China, Beijing, China.

## Impact Statement

This paper presents work whose goal is to advance the field of Machine Learning. There are many potential societal consequences of our work, none which we feel must be specifically highlighted here.

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

# A. Appendix

## A.1. Auxiliary Efficiency Measurements

We additionally measured the per-sample token consumption, wall-clock runtime, and API cost of representative verification settings. All methods were executed with the same concurrency, so the wall-clock runtime is affected by implementation and serving conditions but remains largely aligned with total token consumption. We therefore report equivalent output tokens as the primary hardware-agnostic efficiency metric, while runtime and API cost provide concrete reference points for practical deployment.

*Table 4.* Auxiliary efficiency measurements on representative verification settings. The token usage is measured by equivalent output tokens. Runtime and cost are reported per sample under the same parallel execution setting.

| Model | Method | Bal. F1 | Eq. Tokens | Time (s) | Cost ($) |
|---|---|---|---|---|---|
| gpt-5-mini | standard | 0.78 | 2,975.5 | 6.15 | 0.0060 |
| gpt-5-mini | pes@12 | 0.87 | 35,675.8 | 78.55 | 0.0714 |
| gpt-5-mini | prog@3/6 | 0.85 | 7,146.6 | 14.77 | 0.0143 |
| qwen3-30b-a3b | standard | 0.52 | 3,089.1 | 12.59 | 0.0013 |
| qwen3-30b-a3b | pes@12 | 0.77 | 40,666.5 | 126.95 | 0.0175 |
| qwen3-30b-a3b | prog@3/6 | 0.79 | 11,680.1 | 31.17 | 0.0049 |

These auxiliary measurements are consistent with the efficiency trends in Figure 4. At comparable performance, progressive pessimistic verification uses roughly one fourth of the equivalent tokens, runtime, and API cost of the simple pessimistic verification setting commonly used in agentic math workflows.

## A.2. Details in QiuZhen-Bench

QiuZhen-Bench is a dataset covering a wide range of topics in advanced math, including analysis, algebra, machine learning theory, geometry, topology, probability, statistics and theoretical physics. We have collected 143 exam papers from S.-T. Yau College Student Mathematics Contest since 2010, and the doctoral qualifying exams from QiuZhen college, Tsinghua University since 2023, and required GPT-5 to extract and reformat problems directly from the PDF files. Manual spot checks did not reveal any errors in these extracted contents, indicating the strong capability of GPT-5 in doing this task. We firstly divide the exam papers into pages, and for each page we used the following prompt for extraction:

**system prompt**

```
You are a meticulous exam parsing assistant.
Extract ALL distinct problems from the provided text chunks.
Return ONLY valid JSON (no commentary). Focus on problem statements; exclude answers
    and solutions.
```

**user prompt**

```
Task: Parse the following exam/contest page images to extract problem statements.
Requirements:
- Identify each separate problem with its number or index if present.
- Preserve math using Markdown and LaTeX (inline '$...$', display '$$...$$').
- Do not infer content; only use what appears in the images.
- Only include problems written in English; skip non-English (e.g., Chinese).
- If a problem is mixed-language, include only the English statement; otherwise skip.
- If a problem spans multiple pages in this chunk, include the full statement.
Output strictly as JSON array of objects with keys:
  - problem_number: string (e.g., '1', 'II-3', 'A.1'; use 'unknown' if missing)
  - markdown_statement: string (problem text in Markdown; no solution)
  - section: string|null (e.g., 'Analysis', 'Geometry', or null)
  - tags: array of strings (optional keywords, may be empty)
```

```
   - source_pages: array of integers (page numbers within this PDF chunk)
Return ONLY JSON.

fSource: {source.rel_path}
Pages: {chunk.start_page}-{chunk.end_page}
```

The problems that spans across two pages is extracted by part and combined in the following processing logic. This results in a dataset containing 1,054 problems in total. Here we provide some samples from this dataset:

**Yau contest 2015, applied math**

Let $G$ be graph of a social network, where for each pair of members there is either no connection, or a positive or a negative one. An unbalanced cycle in $G$ is a cycle which have odd number of negative edges. Traversing along such a cycle with social rules such as friend of enemy are enemy would result in having a negative relation of one with himself! A resigning in $G$ at a vertex $v$ of $G$ is to switch the type (positive or negative) of all edges incident to $v$. Question: Show that one can switch all edge of $G$ into positive edges using a sequence resigning if and only if there is no unbalanced cycle in $G$.

**QiuZhen qualifying 2024 spring, theoretical physics**

$\Pi_{\mu\nu}(q)$ via dimensional regularization (35 points)
At one-loop, the photon propagator in QED receives the correction

$$i\,\Pi_{\mu\nu}(q) = -\int \frac{d^4 p}{(2\pi)^4}\,\mathrm{Tr}\left(i\,\gamma^\nu\,\frac{i}{\not{p}+\not{q}-m}\,i\,\gamma^\mu\,\frac{i}{\not{p}-m}\right),$$

where $\gamma_\mu$ is the gamma matrix and $\not{p} = p^\mu \gamma_\mu$.
i) 10' Show that this expression can be written as

$$i\,\Pi_{\mu\nu}(q) = -i\int \frac{d^4 p}{(2\pi)^4}\,\frac{N_{\mu\nu}}{D},$$

where

$$N_{\mu\nu} = \mathrm{tr}\left[\gamma_\nu(\not{p}+\not{q}+m)\gamma_\mu(\not{p}-m)\right], \qquad \frac{1}{D} = \int_0^1 d\alpha\,\frac{1}{D},$$

with

$$D = \left[l^2 + \alpha(1-\alpha)q^2 - m^2 + i\epsilon\right]^2, \qquad \text{where } l = p + \alpha q.$$

ii) 10' Evaluate $N_{\mu\nu}$ and shift momentum integration from $p$ to $l$ as above.
iii) 10' At the integration step over $l$, perform dimensional regularization by promoting to a $d$-dimensional integral. Derive the expression

$$\Pi_{\mu\nu}(q) = (q_\mu q_\nu - \eta_{\mu\nu}q^2)\,\Pi(q^2),$$

when taking the limit $d \to 4$.
iv) 5' Why does this result ensure gauge invariance?

**Yau contest 2019, analysis individual**

Let $\Omega \subset \mathbb{R}^2$ be a bounded domain with smooth boundary. Prove that, for all $p > 1$ and $1 \le q < \infty$, for all $f \in L^p(\Omega)$, there exists a unique $u \in H_0^1(\Omega)$, such that

$$\Delta u = |u|^{q-1}u + f \quad \text{in } \Omega.$$

---

**QiuZhen qualifying 2025 fall, algebra**

Suppose $r \leq n$ are positive integers. Let $X$ be the subset of $M(n \times n, \mathbb{C})$ consisting of complex $n \times n$ matrices with rank at most $r$. Prove that $X$ is Zariski closed. Find the number of irreducible components of $X$, and calculate the Krull dimension of each irreducible component of $X$.

---

## A.3. Prompt Template

In our experiments, we use the same evaluation prompt template across all these datasets. For standard, majority voting, and simple pessimistic verification, we use this single-pass verification prompt:

---

**system prompt: single pass**

```
You are an assistant highly proficient in mathematics.
The user will provide a math problem together with its proposed solution, and your
    task is to verify the correctness of that solution according to the given
    instruction.
```

---

**user prompt: single pass**

```
Here is a math problem and a candidate solution of it, and you need to verify the
    correctness of this solution. Please check each of the following:

1. The provided content is indeed a math problem and its corresponding solution,
    rather than unrelated material supplied by mistake.
2. The solution actually derives the conclusion required by the original problem.
3. Every step of calculation and formula derivation in the solution is correct.
4. The hypotheses (conditions) and conclusions of any theorems used are correctly
    matched and applied.
5. The solution relies only on the conditions given in the problem and does not
    introduce any additional assumptions to obtain the conclusion.

Consistency and error-severity policy (important):
- If only minor, easily fixable issues exist (e.g., small algebraic slips later
    corrected, notational typos, superficial formatting), treat the solution as
    correct but briefly note such issues.
- If there is any critical error that undermines correctness (e.g., invalid step,
    wrong theorem usage without required conditions, uncorrected calculation error
    leading to a wrong result), treat the solution as incorrect.

Response requirements: If the solution is correct overall (possibly with minor issues)
    , reply with '<verification>true</verification>' and briefly list minor issues if
    any.
 If the solution is incorrect, reply with '<verification>false</verification>'
    followed by a concise description of the most harmful error.
 Do not include any restatement of the entire solution or problem.

<problem>{problem}</problem>

<answer>{solution}</answer>
```

---

The first step of progressive pessimistic verifier also adopts the single pass prompt, and for the following verification of subdivisions, we will use the following chunk verification prompt. This prompt is also used in vertical pessimistic verification in our experiments.

```
You are an assistant highly proficient in mathematics.
The user will provide a math problem together with its proposed solution, and your
    task is to verify the correctness of that solution according to the given
    instruction.
```

**user prompt: chunk verification**

```
We provide the original problem and the complete proposed solution for full context.
Then we provide a specific chunk from the solution for focused checking.
Your task: Check ONLY the given chunk for errors while considering the overall
    context.

Checklist:
1. The chunk's reasoning and calculations adhere to mathematical correctness.
2. Any theorems used in the chunk match their hypotheses and conclusions.
3. The chunk does not rely on assumptions not justified by the problem or earlier
    proven steps.

Consistency and error-severity policy (important):
- If only minor, easily fixable issues exist (e.g., small algebraic slips later
    corrected, notational typos, superficial formatting), treat the chunk as correct
    overall but briefly note such issues.
- If there is any critical error that undermines correctness in this chunk (e.g.,
    invalid step, wrong theorem usage without required conditions), treat the chunk as
    incorrect.

Response requirements: If the chunk is correct overall (possibly with minor issues),
    reply with '<verification>true</verification>' and briefly list minor issues if
    any.
If the chunk is incorrect, reply with '<verification>false</verification>' followed by
    a concise description of the most harmful error in the proof that you found in
    the chunk.

<problem>{problem}</problem>

<full_answer>{full_proof}</full_answer>

<chunk_index>{idx}</chunk_index>
<chunk>{chunk}</chunk>
```

## A.4. More Detailed Case Studies

In this section we primarily focus on the false negative cases reported by different verifiers, these are the most informative cases in our experiments.

### A.4.1. FALSE NEGATIVE EXAMPLES ON IMO-GRADINGBENCH

**Case 1: problem**

Let $\angle XYZ$ be an acute angle with $\angle XYZ \neq 60°$, and let $A$ be a point inside $\angle XYZ$. Prove that there exists $D \neq A$ inside $\angle XYZ$ and $\theta \in (0, 2\pi)$ satisfying the following condition:
For points $B$ and $C$ on the rays $\overrightarrow{YX}$ and $\overrightarrow{YZ}$ respectively, then

$$\angle BAC = \angle XYZ \implies \angle BDC = \theta.$$

> **Case 1: proof**

**1. Summary**

**a. Verdict:** I have successfully solved the problem. I have rigorously proven that the statement is true.

**b. Method Sketch:** The proof utilizes complex numbers to model the geometric configuration.

1. **Setup**: We establish a coordinate system with $Y$ at the origin. Let $\alpha = \angle XYZ$. The rays $\overrightarrow{YX}$ and $\overrightarrow{YZ}$ correspond to the positive real axis and the ray at angle $\alpha$, respectively. $A = a$ is inside the angle. $B = b$ and $C = ce^{i\alpha}$ ($b > 0, c > 0$). 2. **Analysis of Configurations ($\mathcal{S}_A$)**: We analyze the condition $\angle BAC = \alpha$. This geometric condition implies the directed angle from $\overrightarrow{AB}$ to $\overrightarrow{AC}$ is either $\alpha$ (CCW) or $-\alpha$ (CW). * We prove that the set of CCW configurations is empty when $A$ is strictly inside the angle. * We prove that the set of CW configurations $\mathcal{S}_A$ is non-empty and infinite, corresponding to an interval of values for $b$. We rigorously derive the relationship between $b$ and $c$ (via an intermediate variable $k$) as a rational function. 3. **Constant Angle Condition**: We assume $D = d$ is a point inside the angle such that $\angle BDC = \theta$ is constant for all configurations in $\mathcal{S}_A$. This implies the complex ratio $X(b) = \frac{C(b)-d}{B(b)-d}$ has a constant argument as $b$ varies. 4. **Analysis of the Rational Function**: $X(b)$ is shown to be a rational function $P(b)/Q(b)$, where $\deg(P) = 1$ and $\deg(Q) = 2$. Since $\arg(X(b))$ is constant, $X(b)$ must be proportional to a real rational function. 5. **Poles Analysis**: The poles of $Q(b)$ are $d$ (non-real) and $b_0$ (real). For the corresponding real rational function to exist, its poles must be symmetric under conjugation. This forces a cancellation between $P(b)$ and $Q(b)$. 6. **Case Analysis of Common Roots**: We analyze the possibilities for the common root. * If the common root is $b_0$, it implies $A$ is on the ray $YX$, contradicting the premise. * If the common root is $d$, it leads to two solutions for $D$: $D = A$ and a unique other point $D = D^*$. 7. **Analysis of the Solution $D^*$**: We explicitly calculate $D^*$ and rigorously prove that $D^*$ is always strictly inside the angle $\angle XYZ$. 8. **Exclusion of $D^* = A$**: We determine the conditions under which $D^* = A$. This occurs if and only if $\alpha = 60°$ and $A$ lies on the angle bisector. 9. **Conclusion**: Since the problem stipulates that $\alpha \neq 60°$, we must have $D^* \neq A$. Thus, $D = D^*$ is the required point inside the angle, different from $A$. The constant angle is calculated to be $\theta = \pi - 2\alpha$.

**2. Detailed Solution**

Step 1: Setup and Coordinate System. Let $\alpha = \angle XYZ$. We are given $0 < \alpha < \pi/2$ and $\alpha \neq \pi/3$. We set up a coordinate system in the complex plane with $Y$ at the origin $0$. The ray $\overrightarrow{YX}$ is the positive real axis, and $\overrightarrow{YZ}$ is the ray at angle $\alpha$. Let $A$ be represented by the complex number $a$. Since $A$ is strictly inside the angle, $a = re^{i\phi_1}$ with $r > 0$ and $0 < \phi_1 < \alpha$. Let $B$ and $C$ be represented by $b$ and $ce^{i\alpha}$ respectively, with $b > 0, c > 0$.

Step 2: Analysis of the configurations $\mathcal{S}_A$. The condition is $\angle BAC = \alpha$. This geometric angle condition means the directed angle from $\overrightarrow{AB}$ (represented by $B - A$) to $\overrightarrow{AC}$ (represented by $C - A$) is either $\alpha$ (CCW orientation) or $-\alpha$ (CW orientation).

Case 2.1: CCW orientation. The directed angle is $\alpha$.

$$\frac{ce^{i\alpha} - a}{b - a} = ke^{i\alpha}, \quad \text{for some } k > 0.$$

$ce^{i\alpha} - a = k(b-a)e^{i\alpha}$. Dividing by $e^{i\alpha}$: $c - ae^{-i\alpha} = k(b-a)$. Since $c, k, b$ are real, we equate the imaginary parts: $Im(c - ae^{-i\alpha}) = kIm(b - a)$. $Im(-ae^{-i\alpha}) = kIm(-a)$. $Im(-re^{i(\phi_1-\alpha)}) = kIm(-re^{i\phi_1})$. $-r\sin(\phi_1 - \alpha) = k(-r\sin\phi_1)$. $\sin(\phi_1 - \alpha) = k\sin\phi_1$. Since $0 < \phi_1 < \alpha < \pi/2$, we have $\sin\phi_1 > 0$. Also, $-\pi/2 < -\alpha < \phi_1 - \alpha < 0$. Thus $\sin(\phi_1 - \alpha) < 0$. This implies $k = \frac{\sin(\phi_1-\alpha)}{\sin\phi_1} < 0$. This contradicts the requirement $k > 0$. Therefore, the set of CCW configurations is empty.

Case 2.2: CW orientation. The directed angle is $-\alpha$.

$$\frac{ce^{i\alpha} - a}{b - a} = ke^{-i\alpha}, \quad \text{for some } k > 0.$$

$ce^{i\alpha} - a = k(b-a)e^{-i\alpha}$. Dividing by $e^{i\alpha}$ is not helpful. Let's rearrange to isolate $c$. $ce^{i\alpha} = a + k(b-a)e^{-i\alpha}$. $c = ae^{-i\alpha} + k(b-a)e^{-i2\alpha}$. Since $c$ is real, we equate the imaginary part to zero. $Im(ae^{-i\alpha}) + kIm((b-a)e^{-i2\alpha}) = 0$. $Im(re^{i(\phi_1-\alpha)}) + kIm(be^{-i2\alpha} - re^{i(\phi_1-2\alpha)}) = 0$. $r\sin(\phi_1 - \alpha) + k(b\sin(-2\alpha) - r\sin(\phi_1 - 2\alpha)) = 0$. $r\sin(\phi_1 - \alpha) = k(-b\sin(-2\alpha) + r\sin(\phi_1 - 2\alpha))$. $r\sin(\phi_1 - \alpha) = k(b\sin 2\alpha - r\sin(\phi_1 - 2\alpha))$.
We rewrite this using $\sin(x) = -\sin(-x)$. $-r\sin(\alpha - \phi_1) = k(b\sin 2\alpha + r\sin(2\alpha - \phi_1))$.
Let $C_1 = r\sin(\alpha - \phi_1)$. Since $0 < \alpha - \phi_1 < \pi/2$, $C_1 > 0$. Let $C_2 = \sin 2\alpha$. Since $0 < 2\alpha < \pi$, $C_2 > 0$. Let $C_3 = r\sin(2\alpha - \phi_1)$. Since $0 < \phi_1 < \alpha$ and $\alpha < \pi/2$, we have $\alpha < 2\alpha - \phi_1 < 2\alpha < \pi$. Thus $C_3 > 0$.

The equation is $-C_1 = k(C_2 b + C_3)$. Since $C_1 > 0$ and $C_2, C_3, b > 0$, the RHS is $k(C_2 b + C_3)$. If $k > 0$, the RHS is positive, while the LHS is negative. This is a contradiction.

Let's recheck the derivation in the thought block, as I seem to have made a mistake again in the live derivation attempt. Let's restart the derivation for Case 2.2 carefully. $ce^{i\alpha} - a = k(b - a)e^{-i\alpha}$. We want to eliminate $c$. We can multiply by $e^{-i\alpha}$. $c - ae^{-i\alpha} = k(b - a)e^{-i2\alpha}$. Since $c$ is real, the imaginary part of the LHS is $Im(-ae^{-i\alpha})$. The imaginary part of the RHS is $kIm((b - a)e^{-i2\alpha})$. $Im(-re^{i(\phi_1 - \alpha)}) = kIm((b - re^{i\phi_1})e^{-i2\alpha})$. $-r\sin(\phi_1 - \alpha) = kIm(be^{-i2\alpha} - re^{i(\phi_1 - 2\alpha)})$. $-r\sin(\phi_1 - \alpha) = k(b\sin(-2\alpha) - r\sin(\phi_1 - 2\alpha))$.

Now we simplify. LHS: $r\sin(\alpha - \phi_1)$. RHS: $k(-b\sin 2\alpha + r\sin(2\alpha - \phi_1))$.

The equation is $r\sin(\alpha - \phi_1) = k(r\sin(2\alpha - \phi_1) - b\sin 2\alpha)$.

Using the definitions of $C_1, C_2, C_3$: $C_1 = k(C_3 - C_2 b)$.

We require $k > 0$. Since $C_1 > 0$, we need $C_3 - C_2 b > 0$. $b < C_3/C_2$. Let $b_0 = C_3/C_2$. $b_0 > 0$. $k(b) = \frac{C_1}{C_3 - C_2 b}$.

We also require $c(b) > 0$. $c(b) = Re(ae^{-i\alpha}) + k(b)Re((b - a)e^{-i2\alpha})$. $c(b) = \frac{Re(ae^{-i\alpha})(C_3 - C_2 b) + C_1 Re((b - a)e^{-i2\alpha})}{C_3 - C_2 b}$.

The denominator is positive for $b \in (0, b_0)$. We analyze the numerator $N(b) = L_1 b + L_0$. $L_1 = -C_2 Re(ae^{-i\alpha}) + C_1 Re(e^{-i2\alpha})$. $L_1 = -\sin 2\alpha \cdot r\cos(\phi_1 - \alpha) + r\sin(\alpha - \phi_1)\cos 2\alpha$. $L_1 = -r[\sin 2\alpha \cos(\alpha - \phi_1) - \cos 2\alpha \sin(\alpha - \phi_1)]$. $L_1 = -r\sin(2\alpha - (\alpha - \phi_1)) = -r\sin(\alpha + \phi_1)$. Since $0 < \alpha + \phi_1 < \pi$, $L_1 < 0$.

$L_0 = C_3 Re(ae^{-i\alpha}) + C_1 Re(-ae^{-i2\alpha})$. $L_0 = r\sin(2\alpha - \phi_1)r\cos(\alpha - \phi_1) - r\sin(\alpha - \phi_1)r\cos(2\alpha - \phi_1)$. $L_0 = r^2 \sin((2\alpha - \phi_1) - (\alpha - \phi_1)) = r^2 \sin \alpha$. Since $\alpha > 0$, $L_0 > 0$.

We need $N(b) > 0$. $L_1 b + L_0 > 0$. Since $L_1 < 0$, $b < -L_0/L_1$. Let $b_1 = -L_0/L_1$. $b_1 > 0$. The set of valid configurations $\mathcal{S}_A$ corresponds to the interval $I_A = (0, \min(b_0, b_1))$. This interval is non-empty, so $\mathcal{S}_A$ is infinite.

Step 3: Analysis of the constant angle condition. Let $D = d$ be a point inside the angle. This means $d$ is not real and $0 < \arg(d) < \alpha$. We require $D \neq A$. We require $\angle BDC = \theta$ to be constant for all $(B, C) \in \mathcal{S}_A$. Let $C(b)$ denote the complex coordinate of $C$ corresponding to $B = b$. $C(b) = c(b)e^{i\alpha}$. The condition means the argument of the complex ratio $X(b) = \frac{C(b) - d}{b - d}$ is constant for $b \in I_A$.

From the definition of the CW configuration in Step 2.2, we have $C(b) = a + k(b)(b - a)e^{-i\alpha}$. $X(b) = \frac{a - d + k(b)(b - a)e^{-i\alpha}}{b - d}$. Substituting $k(b) = \frac{C_1}{C_3 - C_2 b}$: $X(b) = \frac{(a - d)(C_3 - C_2 b) + C_1(b - a)e^{-i\alpha}}{(b - d)(C_3 - C_2 b)} = \frac{P(b)}{Q(b)}$.

$P(b)$ is a polynomial of degree at most 1 in $b$. $Q(b)$ is a polynomial of degree 2 in $b$. $Q(b) = -C_2(b - d)(b - b_0)$, where $b_0 = C_3/C_2$. The roots are $d$ (non-real) and $b_0$ (real and positive).

Since $\arg(X(b))$ is constant on the interval $I_A$, $X(b) = KR(b)$ where $K$ is a complex constant and $R(b)$ is a real rational function. The poles of a real rational function must be real or come in conjugate pairs.

If $P(b)$ and $Q(b)$ are coprime, the poles of $X(b)$ (and thus $R(b)$) are $d$ and $b_0$. Since $d$ is not real, $\bar{d}$ must also be a pole. So $\bar{d} \in \{d, b_0\}$. $\bar{d} = d$ is impossible as $d$ is not real. $\bar{d} = b_0$. Since $b_0$ is real, $d = \bar{b}_0 = b_0$. This implies $d$ is real, which contradicts that $D$ is inside the angle. Thus, $P(b)$ and $Q(b)$ must share a common root.

Step 4: Case analysis of the common root. Case 4.1: The common root is $b_0$. $P(b_0) = 0$. $P(b_0) = (a - d)(C_3 - C_2 b_0) + C_1(b_0 - a)e^{-i\alpha}$. Since $C_3 - C_2 b_0 = 0$, $P(b_0) = C_1(b_0 - a)e^{-i\alpha}$. $P(b_0) = 0$ implies $C_1(b_0 - a)e^{-i\alpha} = 0$. Since $C_1 > 0$ and $e^{-i\alpha} \neq 0$, we must have $b_0 - a = 0$. $a = b_0$. This means $A$ is real, so $A$ is on the ray $YX$. This contradicts that $A$ is strictly inside the angle ($0 < \phi_1$).

Case 4.2: The common root is $d$. $P(d) = 0$. $P(d) = (a - d)(C_3 - C_2 d) + C_1(d - a)e^{-i\alpha} = 0$. $(a - d)[(C_3 - C_2 d) - C_1 e^{-i\alpha}] = 0$. This yields two possibilities for $D$: $D = A$ (i.e., $d = a$) or $C_3 - C_2 d - C_1 e^{-i\alpha} = 0$.

We are looking for $D \neq A$. Let $D^*$ be the second solution, with coordinate $d^*$. $C_2 d^* = C_3 - C_1 e^{-i\alpha}$. $d^* = \frac{C_3 - C_1 e^{-i\alpha}}{C_2}$.

If $D = D^*$, then $(b - d^*)$ is a common factor of $P(b)$ and $Q(b)$. $P(b) = p_1(b - d^*)$. $Q(b) = -C_2(b - d^*)(b - b_0)$. $X(b) = \frac{p_1}{-C_2(b - b_0)}$. For $b \in I_A$, $b < b_0$. Since $C_2 > 0$, the denominator $-C_2(b - b_0)$ is positive and real. Thus, $\arg(X(b)) = \arg(p_1)$, which is constant.

Step 5: Analysis of the location of $D^*$. We must verify that $D^*$ is inside the angle, i.e., $0 < \arg(d^*) < \alpha$. $Im(d^*) = \frac{1}{C_2}Im(C_3 - C_1 e^{-i\alpha}) = \frac{1}{C_2}(-C_1 \sin(-\alpha)) = \frac{C_1 \sin \alpha}{C_2}$. Since $C_1 > 0, C_2 > 0, \sin \alpha > 0$, we have $Im(d^*) > 0$. So $\arg(d^*) > 0$.

We check $\arg(d^*) < \alpha$. This is equivalent to $Im(d^* e^{-i\alpha}) < 0$. $d^* e^{-i\alpha} = \frac{C_3 e^{-i\alpha} - C_1 e^{-i2\alpha}}{C_2}$. $Im(d^* e^{-i\alpha}) = \frac{C_3 \sin(-\alpha) - C_1 \sin(-2\alpha)}{C_2} = \frac{-C_3 \sin \alpha + C_1 \sin 2\alpha}{C_2}$. Let $H = C_1 \sin 2\alpha - C_3 \sin \alpha$. $H = r\sin(\alpha - \phi_1)\sin 2\alpha - r\sin(2\alpha - \phi_1)\sin \alpha$. $H = r\sin \alpha[2\sin(\alpha - \phi_1)\cos \alpha - \sin(2\alpha - \phi_1)]$. Using the identity $2\sin A \cos B = \sin(A + B) + \sin(A - B)$: $2\sin(\alpha - \phi_1)\cos \alpha = \sin(2\alpha - \phi_1) + \sin(-\phi_1) = \sin(2\alpha - \phi_1) - \sin \phi_1$. $H = r\sin \alpha[\sin(2\alpha - \phi_1) -$

$\sin \phi_1 - \sin(2\alpha - \phi_1)]$. $H = -r \sin \alpha \sin \phi_1$. Since $r > 0, \sin \alpha > 0, \sin \phi_1 > 0$ (as $0 < \phi_1 < \alpha < \pi/2$), we have $H < 0$. Thus $Im(d^* e^{-i\alpha}) < 0$. So $\arg(d^*) < \alpha$. $D^*$ is strictly inside the angle $\angle XYZ$.

Step 6: Analysis of the condition $D^* = A$. We check when $d^* = a$. This happens if $a$ satisfies the equation for $d^*$. $C_2 a = C_3 - C_1 e^{-i\alpha}$. $re^{i\phi_1} \sin 2\alpha = r \sin(2\alpha - \phi_1) - r \sin(\alpha - \phi_1)e^{-i\alpha}$. Dividing by $r$ and examining the imaginary part: $\sin \phi_1 \sin 2\alpha = Im(-\sin(\alpha - \phi_1)e^{-i\alpha}) = -\sin(\alpha - \phi_1)\sin(-\alpha) = \sin(\alpha - \phi_1)\sin \alpha$. $2 \sin \phi_1 \sin \alpha \cos \alpha = (\sin \alpha \cos \phi_1 - \cos \alpha \sin \phi_1)\sin \alpha$. Since $\sin \alpha \neq 0$: $2 \sin \phi_1 \cos \alpha = \sin \alpha \cos \phi_1 - \cos \alpha \sin \phi_1$. $3 \sin \phi_1 \cos \alpha = \sin \alpha \cos \phi_1$. $3 \tan \phi_1 = \tan \alpha$. (Since $\cos \alpha \neq 0, \cos \phi_1 \neq 0$).

Examining the real part: $\cos \phi_1 \sin 2\alpha = \sin(2\alpha - \phi_1) - \sin(\alpha - \phi_1)\cos \alpha$. $\cos \phi_1 \sin 2\alpha = (\sin 2\alpha \cos \phi_1 - \cos 2\alpha \sin \phi_1) - (\sin \alpha \cos \phi_1 - \cos \alpha \sin \phi_1)\cos \alpha$. $0 = -\cos 2\alpha \sin \phi_1 - \sin \alpha \cos \alpha \cos \phi_1 + \cos^2 \alpha \sin \phi_1$. $0 = \sin \phi_1(\cos^2 \alpha - \cos 2\alpha) - \sin \alpha \cos \alpha \cos \phi_1$. $0 = \sin \phi_1(\cos^2 \alpha - (\cos^2 \alpha - \sin^2 \alpha)) - \frac{1}{2} \sin 2\alpha \cos \phi_1$. $0 = \sin \phi_1 \sin^2 \alpha - \sin \alpha \cos \alpha \cos \phi_1$. $0 = \sin \alpha(\sin \phi_1 \sin \alpha - \cos \alpha \cos \phi_1) = -\sin \alpha \cos(\alpha + \phi_1)$. Since $\sin \alpha \neq 0$, $\cos(\alpha + \phi_1) = 0$. Since $0 < \alpha + \phi_1 < \pi$, we must have $\alpha + \phi_1 = \pi/2$.

If $\alpha + \phi_1 = \pi/2$, then $\tan \alpha = \cot \phi_1 = 1/\tan \phi_1$. Substituting into $3 \tan \phi_1 = \tan \alpha$: $3 \tan \phi_1 = 1/\tan \phi_1$. $\tan^2 \phi_1 = 1/3$. Since $\phi_1 > 0$, $\tan \phi_1 = 1/\sqrt{3}$. $\phi_1 = \pi/6$ (30 degrees). Then $\alpha = \pi/2 - \pi/6 = \pi/3$ (60 degrees). So $D^* = A$ if and only if $\alpha = 60°$ and $A$ is on the angle bisector ($\phi_1 = \alpha/2$).

Step 7: Conclusion. The problem states that $\angle XYZ \neq 60°$ (i.e., $\alpha \neq \pi/3$). Therefore, the condition for $D^* = A$ is not met, so $D^* \neq A$. We have shown that $D^*$ is strictly inside the angle $\angle XYZ$. We have shown that for $D = D^*$, the angle $\angle BDC$ is constant for all configurations $(B, C)$ such that $\angle BAC = \alpha$.

We determine the constant angle $\theta$. The directed angle $\Theta$ is $\arg(X(b)) = \arg(p_1)$. $p_1$ is the leading coefficient of $P(b)$. $p_1 = -C_2(a - d^*) + C_1 e^{-i\alpha}$. Using the definition of $d^*$, $C_2 d^* = C_3 - C_1 e^{-i\alpha}$, so $C_1 e^{-i\alpha} = C_3 - C_2 d^*$. $p_1 = -C_2 a + C_2 d^* + (C_3 - C_2 d^*) = C_3 - C_2 a$. $p_1 = r \sin(2\alpha - \phi_1) - \sin 2\alpha \cdot re^{i\phi_1}$. $p_1 = r[\sin(2\alpha - \phi_1) - (\cos \phi_1 + i \sin \phi_1)\sin 2\alpha]$. $p_1 = r[(\sin 2\alpha \cos \phi_1 - \cos 2\alpha \sin \phi_1) - \sin 2\alpha \cos \phi_1 - i \sin 2\alpha \sin \phi_1]$. $p_1 = r[-\cos 2\alpha \sin \phi_1 - i \sin 2\alpha \sin \phi_1]$. $p_1 = -r \sin \phi_1(\cos 2\alpha + i \sin 2\alpha) = -r \sin \phi_1 e^{i2\alpha}$. Since $r > 0, \sin \phi_1 > 0$, $\arg(p_1) = \pi + 2\alpha$. The directed angle is $\Theta = \pi + 2\alpha \pmod{2\pi}$. Since $0 < \alpha < \pi/2, \pi < \Theta < 2\pi$. The geometric angle $\theta$ is $2\pi - \Theta = 2\pi - (\pi + 2\alpha) = \pi - 2\alpha$. Since $0 < \alpha < \pi/2, 0 < \theta < \pi$. Thus $\theta \in (0, 2\pi)$.

We have proven the existence of $D = D^* \neq A$ inside $\angle XYZ$ and $\theta = \pi - 2\alpha \in (0, 2\pi)$ satisfying the condition.

---

**Case 1: review from gpt-5**

`<verification>false</verification>` The bilinear relation from $Im\left(\frac{(c-a)e^{-i\alpha}}{be^{i\alpha}-a}\right) = 0$ is computed with a sign error. Expanding $Im(N \cdot \overline{D}) = 0$ gives $cb \sin(2\alpha) + rb \sin(2\alpha - \phi) - rc \sin(\alpha + \phi) + r^2 \sin \alpha = 0$, so the coefficient of $b$ is $A_{12} = r \sin(2\alpha - \phi)$, not $A_{12} = -r \sin(2\alpha - \phi)$ as stated. This sign error undermines the subsequent locus equation.

---

The manually annotated label in the dataset is true (7 points). However, our verification method found an algebraic error in the proof's procedure, suggesting the score should be lower (e.g., 5-6 points).

**Case 1: our comment**

The derivation of the bilinear relation coefficients ($A_{11}$ through $A_{22}$) contained a systematic sign error. For instance, the coefficient $A_{12}$ was calculated with the wrong sign. This error did not propagate to the final construction of point $D$ because the necessary values were derived from ratios of the coefficients (e.g., $A_{22}/A_{11}$), where the overall sign difference cancelled out.

**Case 2: problem**

Find all functions $f : \mathbb{Z} \to \mathbb{Z}$ such that the equation

$$f(x - f(xy)) = f(x)f(1 - y)$$

holds for all $x, y \in \mathbb{Z}$.

## Case 2: proof

We are looking for all functions $f : \mathbb{Z} \to \mathbb{Z}$ such that the equation

$$f(x - f(xy)) = f(x)f(1 - y) \quad (*)$$

holds for all $x, y \in \mathbb{Z}$. Let $P(x, y)$ be the assertion $(*)$.

**1. Constant solutions** If $f(x) = c$ for all $x \in \mathbb{Z}$. Then $c = c \cdot c$, so $c^2 = c$. This implies $c = 0$ or $c = 1$. Solution 1: $f(x) = 0$. (LHS=0, RHS=0). Solution 2: $f(x) = 1$. (LHS=1, RHS=1).

**2. Non-constant solutions** $P(0, y) \implies f(0 - f(0)) = f(0)f(1 - y)$. $f(-f(0)) = f(0)f(1 - y)$. The LHS is constant. If $f(0) \neq 0$, then $f(1 - y)$ is constant. As $y$ spans $\mathbb{Z}$, $f(x)$ is constant. This contradicts the assumption that $f$ is non-constant. Thus, we must have $f(0) = 0$.

$P(x, 0) \implies f(x - f(0)) = f(x)f(1)$. $f(x) = f(x)f(1)$. Since $f$ is not identically zero, there exists $x_0$ such that $f(x_0) \neq 0$. Thus, $f(1) = 1$.

$P(x, 1) \implies f(x - f(x)) = f(x)f(1 - 1) = f(x)f(0) = 0$. (E1) $P(1, y) \implies f(1 - f(y)) = f(1)f(1 - y) = f(1 - y)$. (E2)

Let $Z = \{z \in \mathbb{Z} \mid f(z) = 0\}$ be the set of zeros of $f$. We know $0 \in Z$. (E1) implies that $x - f(x) \in Z$ for all $x \in \mathbb{Z}$.

**3. Case $Z = \{0\}$** If $0$ is the only zero, then $x - f(x) = 0$ for all $x$. Solution 3: $f(x) = x$. Verification: LHS $= f(x - xy) = x - xy$. RHS $= f(x)f(1 - y) = x(1 - y) = x - xy$.

**4. Case $Z \neq \{0\}$** Let $k$ be the smallest positive integer in $Z$. Since $f(1) = 1$, $k \geq 2$.

If $z \in Z$, $f(z) = 0$. By (E2), $f(1 - z) = f(1 - f(z)) = f(1 - 0) = f(1) = 1$.

We will show that $k$ must be 2 or 3.

**Step 4.1: Suppose $k$ is even.** Let $k = 2m$ for some integer $m \geq 1$. $f(k) = f(2m) = 0$. $P(m, 2) \implies f(m - f(2m)) = f(m)f(1 - 2)$. $f(m - 0) = f(m)f(-1)$. $f(m) = f(m)f(-1)$.

If $m = 1$, then $k = 2$. If $m > 1$, then $1 < m < k$. By the definition of $k$, $f(m) \neq 0$. Thus $f(-1) = 1$. $P(1, -1) \implies f(1 - f(-1)) = f(1)f(1 - (-1))$. $f(1 - 1) = 1 \cdot f(2)$. $f(0) = f(2)$. So $f(2) = 0$. This means $k \leq 2$. Since $k$ is even, $k = 2$. So, if $k$ is even, $k = 2$.

**Step 4.2: Suppose $k$ is odd.** $k \geq 3$. $f(k) = 0$. Since $k$ is odd, $y = \frac{1-k}{2}$ is an integer. $1 - y = \frac{1+k}{2}$. $P(2, y) \implies P(2, \frac{1-k}{2})$. $f(2 - f(2y)) = f(2)f(1 - y)$. $f(2 - f(1 - k)) = f(2)f(\frac{1+k}{2})$. We know $f(1 - k) = 1$ (since $k \in Z$). $f(2 - 1) = f(1) = 1$. So $1 = f(2)f(\frac{1+k}{2})$.

Since $f(x) \in \mathbb{Z}$, $f(2)$ must be 1 or $-1$.

If $f(2) = 1$. By (E1), $f(2 - f(2)) = f(2 - 1) = f(1) = 0$. But $f(1) = 1$. Contradiction ($1 = 0$). If $f(2) = -1$. By (E1), $f(2 - f(2)) = f(2 - (-1)) = f(3) = 0$. This implies $k \leq 3$. Since $k$ is odd and $k \geq 3$, we must have $k = 3$. So, the smallest positive zero $k$ can only be 2 or 3.

**Step 4.3: Case k=2.** $f(2) = 0$. $f(1) = 1$. We look for a solution where $Z = 2\mathbb{Z}$ (the set of even integers). If $Z = 2\mathbb{Z}$, $f(x) = 0$ iff $x$ is even. Let $x$ be odd (so $f(x) \neq 0$) and $y$ be even. $xy$ is even, so $f(xy) = 0$. $1 - y$ is odd. $P(x, y) \implies f(x - f(xy)) = f(x)f(1 - y)$. $f(x - 0) = f(x)f(1 - y)$. $f(x) = f(x)f(1 - y)$. Since $f(x) \neq 0$, we must have $f(1 - y) = 1$. As $y$ ranges over all even integers, $1 - y$ ranges over all odd integers. So $f(z) = 1$ for all odd $z$.

Solution 4: $f(x) = \begin{cases} 0 & \text{if } x \text{ is even} \\ 1 & \text{if } x \text{ is odd} \end{cases}$ (i.e., $f(x) = x \bmod 2$). Verification: Case 1: $x$ is even. $f(x) = 0$. RHS=0.

$f(xy)$ is 0 (if $y$ is even) or $f(x) = 0$ (if $y$ is odd). In both cases $f(xy) = 0$. LHS=$f(x - 0) = 0$. Case 2: $x$ is odd. $f(x) = 1$. RHS=$f(1 - y)$. If $y$ is even, $1 - y$ is odd (RHS=1). $xy$ is even, $f(xy) = 0$. LHS=$f(x - 0) = f(x) = 1$. If $y$ is odd, $1 - y$ is even (RHS=0). $xy$ is odd, $f(xy) = 1$. LHS=$f(x - 1)$. Since $x$ is odd, $x - 1$ is even, LHS=0.

**Step 4.4: Case k=3.** $f(3) = 0$. $f(1) = 1$. We found $f(2) = -1$. We look for a solution where $Z = 3\mathbb{Z}$. $x - f(x) \in Z = 3\mathbb{Z}$, so $f(x) \equiv x \pmod 3$. If $f(x) \in \{-1, 0, 1\}$, then $f(x)$ is uniquely determined by $x \bmod 3$.

Solution 5: $f(x) = \begin{cases} 0 & \text{if } x \equiv 0 \pmod 3 \\ 1 & \text{if } x \equiv 1 \pmod 3 \\ -1 & \text{if } x \equiv 2 \pmod 3 \end{cases}$. (This is the Legendre symbol $(x/3)$). Verification: This function $f$ is

completely multiplicative, $f(xy) = f(x)f(y)$, and $f(x) \in \{-1, 0, 1\}$. The equation is $f(x - f(x)f(y)) = f(x)f(1 - y)$. If $f(x) = 0$ (i.e., $3|x$), LHS=0, RHS=0. If $f(x) \neq 0$, then $f(x) \in \{1, -1\}$. LHS $= f(x(1 - f(x)^{-1}f(y)))$. Since $f(x)^{-1} = f(x)$, LHS $= f(x)f(1 - f(x)f(y))$. We need $f(x)f(1 - f(x)f(y)) = f(x)f(1 - y)$. Since $f(x) \neq 0$, we need $f(1 - f(x)f(y)) = f(1 - y)$.

If $f(x) = 1$. $f(1 - f(y)) = f(1 - y)$. This holds by (E2). If $f(x) = -1$. $f(1 - (-1)f(y)) = f(1 + f(y))$. We need $f(1 + f(y)) = f(1 - y)$. If $f(y) = 0$ ($y \equiv 0 \pmod 3$). $f(1) = 1$. $1 - y \equiv 1 \pmod 3$, $f(1 - y) = 1$. If $f(y) = 1$ ($y \equiv 1 \pmod 3$). $f(1 + 1) = f(2) = -1$. $1 - y \equiv 0 \pmod 3$, $f(1 - y) = 0$. $-1 = 0$.

Let's re-verify Solution 5 using the congruence property $f(x) \equiv x \pmod 3$. LHS $= f(x - f(xy))$. RHS $= f(x)f(1 - y)$. RHS $\equiv x(1 - y) = x - xy \pmod 3$. LHS $\equiv x - f(xy) \pmod 3$. Since $f(xy) \equiv xy \pmod 3$. LHS $\equiv x - xy \pmod 3$. Since LHS and RHS are both in $\{-1, 0, 1\}$ and congruent modulo 3, they must be equal.

Wait, there was an error in the verification attempt using multiplicativity above: If $f(x) = -1$ and $f(y) = 1$. $f(1 + f(y)) = f(2) = -1$. $f(1 - y) = 0$. $-1 = 0$. This derivation must be flawed.

Let's check the specific case: $x = 2$ ($f(x) = -1$), $y = 1$ ($f(y) = 1$). LHS: $f(x - f(xy)) = f(2 - f(2)) = f(2 - (-1)) = f(3) = 0$. RHS: $f(x)f(1 - y) = f(2)f(0) = (-1)(0) = 0$. The specific case holds.

The error in the multiplicative derivation was assuming $f(x - f(x)f(y)) = f(x)f(1 - f(x)f(y))$ when $f(x) \neq 0$. This requires $f(a - b) = f(a)f(1 - b/a)$ which is generally not true. Multiplicativity $f(ab) = f(a)f(b)$ works, but we cannot factor $x$ out if $x$ is not coprime to 3 (and even if it is, $f(x - y) \neq f(x)f(1 - y/x)$).

The verification using $f(x) \equiv x \pmod 3$ is correct.

We have found five solutions.

Final Answer: The functions satisfying the equation are: 1. $f(x) = 0$ for all $x \in \mathbb{Z}$. 2. $f(x) = 1$ for all $x \in \mathbb{Z}$. 3. $f(x) = x$ for all $x \in \mathbb{Z}$. 4. $f(x) = \begin{cases} 0 & \text{if } x \text{ is even} \\ 1 & \text{if } x \text{ is odd} \end{cases}$. 5. $f(x) = \begin{cases} 0 & \text{if } x \equiv 0 \pmod 3 \\ 1 & \text{if } x \equiv 1 \pmod 3 \\ -1 & \text{if } x \equiv 2 \pmod 3 \end{cases}$.

---

**Case 2: review from gpt-5**

The chunk makes critical, unjustified assumptions when classifying the non-constant solutions. In particular:

- In Step 4.3 ($k = 2$), it assumes $Z = 2\mathbb{Z}$ (all even numbers are zeros) and then derives the parity solution, but it never proves from the functional equation that $Z$ must equal $2\mathbb{Z}$ or that $f(\text{odd}) = 1$.

- In Step 4.4 ($k = 3$), it assumes $Z = 3\mathbb{Z}$ and that $f(x) \in \{-1, 0, 1\}$, using this to conclude the "Legendre symbol" form via a congruence argument. These constraints are not derived from $k = 3$ (only $f(x) \equiv x \pmod 3$ is shown).

Thus the claim that the five listed functions are all solutions is not established.

The manually annotated label in the dataset is true (7 points), while our method found a critical error in it:

---

**Case 2: our comment**

The proof is incomplete in steps 4.3 and 4.4 due to unjustified assumptions. For the $k = 2$ and $k = 3$ cases, the author assumes the set of zeros is $Z = k\mathbb{Z}$ and, for $k = 3$, further assumes $f(x) \in \{-1, 0, 1\}$ to match the Legendre symbol form. While these specific functions are indeed solutions, the proof fails to derive that *all* solutions satisfying $f(k) = 0$ must necessarily possess this periodicity and range structure.

---

**Case 3: problem**

For a positive integer $n$, let $A_n$ be the number of perfect powers less than or equal to $n$. Here, a perfect power is a number that can be expressed in the form $a^b$, where $a$ is a positive integer and $b$ is an integer greater than or equal to 2. Prove that there are infinitely many $n$ such that $A_n$ divides $n + 2024$.

---

**Case 3: proof**

Let $S$ be the set of perfect powers. A perfect power is a positive integer that can be expressed in the form $a^b$, where $a$ is a positive integer and $b$ is an integer greater than or equal to 2. The set of perfect powers, sorted in increasing order, is $\{1, 4, 8, 9, 16, 25, 27, \dots\}$. For a positive integer $n$, $A_n$ is the number of perfect powers less than or equal to $n$. So,

$A_n = |S \cap \{1, 2, \ldots, n\}|$. The function $A_n$ is a non-decreasing step function. It is constant between consecutive perfect powers. Let $p_k$ denote the $k$-th perfect power. For any integer $n$ in the range $p_k \leq n < p_{k+1}$, the value of $A_n$ is constant and equal to $k$.

We want to prove that there are infinitely many positive integers $n$ such that $A_n$ divides $n + 2024$.

Our strategy is to find an infinite sequence of disjoint integer intervals, and in each interval, find one such integer $n$. Let's consider an interval $[p_k, p_{k+1} - 1]$ where $A_n$ is constant, equal to $k$. The condition we want to satisfy is $k \mid (n + 2024)$. This is equivalent to the congruence relation $n \equiv -2024 \pmod{k}$. The interval $[p_k, p_{k+1} - 1]$ contains $p_{k+1} - p_k$ consecutive integers. If the length of this interval is at least $k$, the integers it contains form a complete residue system modulo $k$. In such a case, there is guaranteed to be an integer $n$ in the interval satisfying $n \equiv -2024 \pmod{k}$.

So, the problem reduces to showing that there are infinitely many values of $k$ for which the gap between consecutive perfect powers, $p_{k+1} - p_k$, is greater than or equal to $k$.

We will focus on a special subsequence of perfect powers: the squares. Consider the gaps between consecutive squares. Let $m \geq 2$ be an integer. We will analyze the interval of integers $I_m = [(m-1)^2, m^2 - 1]$. We will show that for all sufficiently large $m$: 1. $A_n$ is constant for all $n \in I_m$. To do this, we will prove that there are no perfect powers in the open interval $((m-1)^2, m^2)$. 2. The length of the interval $I_m$ is greater than the value of $A_n$ for $n \in I_m$.

If these two conditions hold for some large $m$, let $k_m$ be the constant value of $A_n$ on $I_m$. The length of $I_m$ is $(m^2 - 1) - (m-1)^2 + 1 = 2m - 1$. Condition 2 means $2m - 1 > k_m$. As argued above, since the interval $I_m$ contains $2m - 1 > k_m$ consecutive integers, there must exist an integer $n_m \in I_m$ such that $n_m \equiv -2024 \pmod{k_m}$. For this $n_m$, we have $A_{n_m} = k_m$, and thus $A_{n_m} \mid (n_m + 2024)$.

If we can show that these conditions hold for all $m$ greater than some integer $M$, we can find a solution $n_m$ for each $m \geq M$. Since the intervals $I_m$ are disjoint for different $m$, the solutions $n_m$ will be distinct, proving that there are infinitely many such integers.

We now formalize and prove these two main claims.

**Claim 1:** For all sufficiently large integers $m$, there are no perfect powers in the open interval $((m-1)^2, m^2)$.

**Proof:** A perfect power in the interval $((m-1)^2, m^2)$ cannot be a perfect square, as there are no integers between $m - 1$ and $m$. Therefore, any such perfect power must be of the form $x^j$ where $x \geq 2$ and $j \geq 3$ are integers. For $x^j$ to be in the interval, we must have $(m-1)^2 < x^j < m^2$. Taking the $j$-th root of this inequality gives $(m-1)^{2/j} < x < m^{2/j}$. For an integer $x$ to exist in this range, the length of the interval $((m-1)^{2/j}, m^{2/j})$ must be greater than 1 for some $j$. The possible values for the exponent $j$ are limited by the condition $2^j \leq x^j < m^2$, which implies $j < 2 \log_2 m$.

Let $N(a, b)$ be the number of perfect powers in the open interval $(a, b)$. The total number of perfect powers in $((m-1)^2, m^2)$ is bounded by the sum of the number of integer candidates for the base $x$ over all possible prime exponents $j$. For simplicity, we sum over all integers $j \geq 3$:

$$N((m-1)^2, m^2) \leq \sum_{j=3}^{\lfloor 2 \log_2 m \rfloor} \left( \lfloor m^{2/j} \rfloor - \lceil (m-1)^{2/j} \rceil + 1 \right)$$

A simpler bound is the sum of the lengths of the intervals for $x$:

$$N((m-1)^2, m^2) \leq \sum_{j=3}^{\lfloor 2 \log_2 m \rfloor} \left( m^{2/j} - (m-1)^{2/j} \right)$$

Let's analyze the term $m^{2/j} - (m-1)^{2/j}$. By the Mean Value Theorem applied to the function $f(t) = t^{2/j}$ on the interval $[m-1, m]$, for each $j$ there exists some $c_j \in (m-1, m)$ such that:

$$m^{2/j} - (m-1)^{2/j} = f'(c_j) \cdot (m - (m-1)) = \frac{2}{j} c_j^{\frac{2}{j} - 1}$$

Since $j \geq 3$, the exponent $\frac{2}{j} - 1$ is negative. As $c_j > m - 1$, we have $c_j^{\frac{2}{j} - 1} < (m-1)^{\frac{2}{j} - 1}$. So, $m^{2/j} - (m-1)^{2/j} < \frac{2}{j}(m-1)^{\frac{2}{j} - 1}$.

Now we bound the sum. For $j \geq 3$, the exponent $\frac{2}{j} - 1 \leq \frac{2}{3} - 1 = -\frac{1}{3}$. Assuming $m - 1 \geq 1$, we have $(m-1)^{\frac{2}{j}-1} \leq (m-1)^{-1/3}$.

$$N((m-1)^2, m^2) < \sum_{j=3}^{\lfloor 2\log_2 m \rfloor} \frac{2}{j}(m-1)^{\frac{2}{j}-1} \leq \sum_{j=3}^{\lfloor 2\log_2 m \rfloor} \frac{2}{3}(m-1)^{-1/3}$$

The number of terms in the summation is $\lfloor 2\log_2 m \rfloor - 2 < 2\log_2 m$.

$$N((m-1)^2, m^2) < (2\log_2 m) \cdot \frac{2}{3}(m-1)^{-1/3} = \frac{4\log_2 m}{3(m-1)^{1/3}}$$

As $m \to \infty$, this upper bound tends to 0. To see this formally:

$$\lim_{m \to \infty} \frac{4\log_2 m}{3(m-1)^{1/3}} = \lim_{m \to \infty} \frac{4\ln m}{3\ln 2 \cdot (m-1)^{1/3}} = 0$$

since any positive power of $m$ grows faster than any power of $\ln m$. Since $N((m-1)^2, m^2)$ must be a non-negative integer, there must exist an integer $M_1$ such that for all $m \geq M_1$, $N((m-1)^2, m^2) = 0$. This proves Claim 1.

**Claim 2:** For all sufficiently large integers $m$, $2m - 1 > A_{(m-1)^2}$.

**Proof:** We need an upper bound for $A_n$. The set of perfect powers is the union of the sets of squares, cubes, 5th powers, and so on. A simple upper bound for $A_n$ is obtained by summing the number of $j$-th powers up to $n$ for all $j \geq 2$, ignoring overlaps which leads to overcounting. The number of $j$-th powers up to $n$ is $\lfloor n^{1/j} \rfloor$.

$$A_n \leq \sum_{j=2}^{\lfloor \log_2 n \rfloor} \lfloor n^{1/j} \rfloor < \sum_{j=2}^{\lfloor \log_2 n \rfloor} n^{1/j}$$

We can write this as:

$$A_n < n^{1/2} + n^{1/3} + n^{1/4} + \cdots + n^{1/\lfloor \log_2 n \rfloor}$$

The number of terms in the sum after $n^{1/2}$ is $\lfloor \log_2 n \rfloor - 2$. For $j \geq 3$, we have $n^{1/j} \leq n^{1/3}$ (for $n \geq 1$). So, we can bound the sum as:

$$A_n < n^{1/2} + (\lfloor \log_2 n \rfloor - 2)n^{1/3} < n^{1/2} + (\log_2 n)n^{1/3}$$

Now, let's apply this bound to $A_{(m-1)^2}$. Let $n = (m-1)^2$.

$$A_{(m-1)^2} < \sqrt{(m-1)^2} + (\log_2((m-1)^2)) \cdot ((m-1)^2)^{1/3}$$

$$A_{(m-1)^2} < m - 1 + (2\log_2(m-1)) \cdot (m-1)^{2/3}$$

We want to prove that $2m - 1 > A_{(m-1)^2}$ for sufficiently large $m$. It is sufficient to show that:

$$2m - 1 > m - 1 + 2\log_2(m-1) \cdot (m-1)^{2/3}$$

This inequality simplifies to:

$$m > 2\log_2(m-1) \cdot (m-1)^{2/3}$$

For $m > 1$, we can divide by $m^{2/3}$:

$$m^{1/3} > 2\log_2(m-1) \cdot \left(\frac{m-1}{m}\right)^{2/3} = 2\log_2(m-1) \cdot \left(1 - \frac{1}{m}\right)^{2/3}$$

As $m \to \infty$, the term $(1 - 1/m)^{2/3}$ approaches 1. The inequality will hold if $m^{1/3}$ grows faster than $2\log_2(m-1)$. To confirm this, we examine the limit of their ratio:

$$\lim_{x \to \infty} \frac{x^{1/3}}{2\log_2(x-1)} = \lim_{x \to \infty} \frac{x^{1/3}\ln 2}{2\ln(x-1)}$$

Using L'Hôpital's rule on this $\frac{\infty}{\infty}$ form:

$$= \lim_{x \to \infty} \frac{\frac{1}{3} x^{-2/3} \ln 2}{2 \cdot \frac{1}{x-1}} = \lim_{x \to \infty} \frac{\ln 2}{6} \frac{x-1}{x^{2/3}} = \lim_{x \to \infty} \frac{\ln 2}{6} (x^{1/3} - x^{-2/3}) = \infty$$

Since this limit is infinite, $x^{1/3}$ grows faster than any constant multiple of $\log_2(x-1)$. Therefore, there exists an integer $M_2$ such that for all $m \geq M_2$, the inequality $m^{1/3} > 2 \log_2(m-1) \cdot (1 - 1/m)^{2/3}$ holds. This proves Claim 2.

**Conclusion**

Let $M = \max(M_1, M_2)$. For every integer $m \geq M$, we have established the following: 1. From Claim 1, the interval $((m-1)^2, m^2)$ contains no perfect powers. This means that for any integer $n$ in the interval $I_m = [(m-1)^2, m^2 - 1]$, the set of perfect powers less than or equal to $n$ is the same as the set of perfect powers less than or equal to $(m-1)^2$. Thus, $A_n$ has a constant value for all $n \in I_m$. Let this value be $k_m = A_{(m-1)^2}$.

2. From Claim 2, the inequality $2m - 1 > A_{(m-1)^2} = k_m$ holds. The length of the interval $I_m$ is $(m^2 - 1) - (m - 1)^2 + 1 = 2m - 1$. So, for $m \geq M$, the length of the interval $I_m$ is strictly greater than $k_m$.

We are looking for an integer $n$ such that $A_n \mid (n + 2024)$. For any $m \geq M$, we seek an $n \in I_m$ satisfying $k_m \mid (n + 2024)$. This is equivalent to finding an $n$ in the range $(m-1)^2 \leq n \leq m^2 - 1$ that satisfies the congruence:

$$n \equiv -2024 \pmod{k_m}$$

The interval $I_m$ contains the sequence of consecutive integers $(m-1)^2, (m-1)^2 + 1, \ldots, m^2 - 1$. The length of this sequence is $2m - 1$. Since $m \geq M$, we have $2m - 1 > k_m$. Because the sequence of integers in $I_m$ is longer than a full cycle of residues modulo $k_m$, it is guaranteed to contain at least one integer from each residue class modulo $k_m$. In particular, there must exist an integer $n_m \in I_m$ such that $n_m \equiv -2024 \pmod{k_m}$. For this $n_m$, we have $A_{n_m} = k_m$, and so $A_{n_m} \mid (n_m + 2024)$.

For each integer $m \geq M$, we have successfully found an integer $n_m$ that satisfies the given property. To complete the proof, we must show that this process generates an infinite number of distinct solutions.

Let $m$ and $m'$ be two integers such that $M \leq m < m'$. The solution $n_m$ lies in the interval $I_m = [(m-1)^2, m^2 - 1]$. The solution $n_{m'}$ lies in the interval $I_{m'} = [(m'-1)^2, (m')^2 - 1]$. Since $m' > m$, we must have $m' \geq m + 1$. The smallest value in the interval $I_{m'}$ is $(m'-1)^2 \geq (m+1-1)^2 = m^2$. The largest value in the interval $I_m$ is $m^2 - 1$. So, for any $n \in I_m$ and $n' \in I_{m'}$, we have $n \leq m^2 - 1 < m^2 \leq n'$. This shows that the intervals $I_m$ and $I_{m'}$ are disjoint. Consequently, the solutions $n_m \in I_m$ and $n_{m'} \in I_{m'}$ are distinct.

Since we can choose any integer $m \geq M$, we have constructed an infinite sequence of distinct solutions $\{n_m\}_{m \geq M}$. Therefore, there are infinitely many integers $n$ such that $A_n$ divides $n + 2024$.

The manually annotated label in the dataset is true (7 points), while our method found a critical error in it:

**Case 3: review from GPT-5-mini**

<verification>false</verification>
The main (fatal) error is in Claim 1: the author replaces the integer count

$$\lfloor m^{2/j} \rfloor - \lceil (m-1)^{2/j} \rceil + 1$$

by the real length $m^{2/j} - (m-1)^{2/j}$. That inequality is not valid in general (e.g. an interval of real length $< 1$ can still contain an integer). Thus the bound

$$N\big((m-1)^2, m^2\big) \leq \sum_j \big( m^{2/j} - (m-1)^{2/j} \big)$$

is unjustified, and the conclusion "this sum $\to 0 \Rightarrow N = 0$ for large $m$" does not follow. Because of this mistake the proof does not establish that the open interval $\big((m-1)^2, m^2\big)$ contains no perfect powers for all large $m$, and the rest of the argument (which depends critically on that claim) fails.

> **Case 4: problem**
>
> Prove that there exist two real-coefficient polynomials $P(x)$ and $Q(x)$ satisfying the following condition:
> (Condition) The degree of the polynomial $P(x)$ is at least 2024, the degree of $Q(x)$ is at least 2, and for any real number $x$, the following holds:
> $$P(Q(x) - x - 1) = Q(P(x))$$

> **Case 4: proof**
>
> We will prove the existence of such polynomials $P(x)$ and $Q(x)$ by constructing them. Our construction will follow three main steps: 1. Find a pair of polynomials of low degree, let's call them $(P_0(x), Q_0(x))$, that satisfies the given functional equation. 2. Construct a high-degree polynomial, let's call it $R(x)$, that commutes with $Q_0(x)$. 3. Combine these polynomials to form the final pair $(P(x), Q(x))$ that satisfies all the given conditions.
> Let the given equation be
> $$P(Q(x) - x - 1) = Q(P(x)) \quad (*).$$
>
> **Step 1: Finding a base solution pair**
> We look for a simple solution where $P_0(x)$ and $Q_0(x)$ have degree 2. Let's try to relate them. A simple assumption is that $Q_0(x)$ is a linear perturbation of $P_0(x)$. Let's try the ansatz:
> $$Q_0(x) = P_0(x) + x + c$$
>
> for some real constant $c$. Substituting this into the equation $(*)$: The left side's argument becomes $Q_0(x) - x - 1 = (P_0(x) + x + c) - x - 1 = P_0(x) + c - 1$. So the equation becomes $P_0(P_0(x) + c - 1) = Q_0(P_0(x))$.
> Let $y = P_0(x)$. For any value $y$ in the range of $P_0(x)$, the equation must hold:
> $$P_0(y + c - 1) = Q_0(y).$$
>
> Using our ansatz for $Q_0(y)$, we have $Q_0(y) = P_0(y) + y + c$. So, we obtain a functional equation for $P_0(y)$:
> $$P_0(y + c - 1) = P_0(y) + y + c.$$
>
> Let $b = c - 1$, so $c = b + 1$. The equation is
> $$P_0(y + b) - P_0(y) = y + b + 1.$$
>
> Let $m = \deg(P_0)$. If $b \neq 0$, the degree of the polynomial on the left, $P_0(y+b) - P_0(y)$, is $m - 1$. The degree of the polynomial on the right is 1. For the identity to hold, the degrees must be equal, so $m - 1 = 1$, which implies $m = 2$. Let $P_0(y) = Ay^2 + By + C$ for some constants $A, B, C \in \mathbb{R}$ with $A$ eq0. Substituting this into the difference equation:
> $A(y+b)^2 + B(y+b) + C - (Ay^2 + By + C) = y + b + 1$ $A(y^2 + 2by + b^2) + B(y+b) + C - = Ay^2 - By - C = y + b + 1$

$2Aby + Ab^2 + Bb = y + b + 1$. This must be a polynomial identity in $y$. We can equate the coefficients of the powers of $y$: 1. Coefficient of $y$: $2Ab = 1$. 2. Constant term: $Ab^2 + Bb = b + 1$.

This is a system of two equations with three unknowns $(A, B, b)$. We can choose a value for one of them. Let's choose $b = 2$. From (1): $2A(2) = 1 \implies 4A = 1 \implies A = 1/4$. Substitute $A = 1/4$ and $b = 2$ into (2): $(1/4)(2^2) + B(2) = 2 + 1 \implies 1 + 2B = 3 \implies 2B = 2 \implies B = 1$. The constant $C$ can be any real number. For simplicity, we choose $C = 0$. So, we have found a polynomial $P_0(x) = \frac{1}{4}x^2 + x$. We have $b = 2$, so $c = b + 1 = 3$.

Now we find the corresponding $Q_0(x)$: $Q_0(x) = P_0(x) + x + c = (\frac{1}{4}x^2 + x) + x + 3 = \frac{1}{4}x^2 + 2x + 3$.

So, we have found a base solution pair: $P_0(x) = \frac{1}{4}x^2 + x$ $Q_0(x) = \frac{1}{4}x^2 + 2x + 3$ Both have degree 2, satisfying the degree condition for $Q_0(x)$. Let's verify that they satisfy $(*)$. $Q_0(x) - x - 1 = (\frac{1}{4}x^2 + 2x + 3) - x - 1 = \frac{1}{4}x^2 + x + 2$.

LHS: $P_0(Q_0(x) - x - 1) = P_0(\frac{1}{4}x^2 + x + 2) = \frac{1}{4}(\frac{1}{4}x^2 + x + 2)^2 + (\frac{1}{4}x^2 + x + 2) = \frac{1}{64}x^4 + \frac{1}{8}x^3 + \frac{3}{4}x^2 + 2x + 3$.

RHS: $Q_0(P_0(x)) = \frac{1}{4}(P_0(x))^2 + 2P_0(x) + 3 = \frac{1}{4}(\frac{1}{4}x^2 + x)^2 + 2(\frac{1}{4}x^2 + x) + 3 = \frac{1}{64}x^4 + \frac{1}{8}x^3 + \frac{3}{4}x^2 + 2x + 3$.

LHS=RHS, so the pair $(P_0, Q_0)$ is a solution.

**Step 2: Construction of a high-degree commuting polynomial**

Our plan is to set $Q(x) = Q_0(x)$ and $P(x) = R(P_0(x))$ for some polynomial $R(x)$ with high degree. Let's see what condition $R(x)$ must satisfy. Substitute $P(x) = R(P_0(x))$ and $Q(x) = Q_0(x)$ into $(*)$: LHS: $P(Q(x) - x - 1) = R(P_0(Q_0(x) - x - 1))$. Using the property of our base solution, $P_0(Q_0(x) - x - 1) = Q_0(P_0(x))$, we get: LHS $= R(Q_0(P_0(x)))$. RHS: $Q(P(x)) = Q_0(R(P_0(x)))$. So, the equation becomes $R(Q_0(P_0(x))) = Q_0(R(P_0(x)))$. Let $y = P_0(x)$. This identity must hold for all values $y$ in the range of $P_0$. If we find a polynomial $R$ that commutes with $Q_0$, i.e., $R(Q_0(y)) = Q_0(R(y))$ for all $y$, then our equation will be satisfied. Since this is a polynomial identity, if it holds for all $y$ in the range of $P_0$ (an infinite set), it holds for all $y$.

We need to find a polynomial $R(x)$ that commutes with $Q_0(x)$. A standard method for finding commuting polynomials is to simplify $Q_0(x)$ by a linear change of variables (conjugation). $Q_0(x) = \frac{1}{4}x^2 + 2x + 3 = \frac{1}{4}(x^2 + 8x) + 3 = \frac{1}{4}(x^2 + 8x + 16) - 4 + 3 = \frac{1}{4}(x + 4)^2 - 1$. Let $L(x) = x + 4$. Its inverse is $L^{-1}(x) = x - 4$. Let $S(y) = \frac{1}{4}y^2 - 1$. Then $Q_0(x) = L^{-1}(S(L(x)))$.

If we find a polynomial $\tilde{R}(y)$ that commutes with $S(y)$, then $R(x) = L^{-1}(\tilde{R}(L(x)))$ will commute with $Q_0(x)$. This is because: $R(Q_0(x)) = L^{-1}(\tilde{R}(L(L^{-1}(S(L(x)))))) = L^{-1}(\tilde{R}(S(L(x))))$. $Q_0(R(x)) = L^{-1}(S(L(L^{-1}(\tilde{R}(L(x)))))) = L^{-1}(S(\tilde{R}(L(x))))$. Since $\tilde{R}$ and $S$ commute, $\tilde{R}(S(y)) = S(\tilde{R}(y))$, so $R$ and $Q_0$ commute.

A simple way to find a polynomial that commutes with $S(y)$ is to take iterates of $S(y)$. Let $S_1(y) = S(y)$ and $S_k(y) = S(S_{k-1}(y))$ for $k \geq 2$. Then $S_k(y)$ commutes with $S(y)$ for any $k \geq 1$. Let's choose $\tilde{R}(y) = S_k(y)$ for a sufficiently large $k$. The degree of $S(y)$ is 2. The degree of $\tilde{R}(y) = S_k(y)$ is $2^k$. We define $R(x) = L^{-1}(S_k(L(x))) = S_k(x + 4) - 4$. The degree of $R(x)$ is $2^k$.

**Step 3: Assembling the final solution**

We are now ready to define $P(x)$ and $Q(x)$. - Let $Q(x) = Q_0(x) = \frac{1}{4}x^2 + 2x + 3$. - Let $R(x)$ be constructed with $k = 10$, so $\tilde{R}(y) = S_{10}(y) = S^{\circ 10}(y)$, and $R(x) = S_{10}(x + 4) - 4$. - Let $P(x) = R(P_0(x)) = R(\frac{1}{4}x^2 + x)$.

Let's check the conditions: 1. **Coefficients**: $P_0, Q_0, L, L^{-1}, S$ are all real-coefficient polynomials. Any composition or iteration will also result in polynomials with real coefficients. So $P(x)$ and $Q(x)$ have real coefficients.

2. **Degree of Q(x)**: $\deg(Q) = \deg(Q_0) = 2$. This satisfies $\deg(Q) \geq 2$.

3. **Degree of P(x)**: The degree of $P(x)$ is the product of the degrees of $R(x)$ and $P_0(x)$. $\deg(P_0) = 2$. $\deg(R) = \deg(S_{10}) = 2^{10} = 1024$. $\deg(P) = \deg(R) \cdot \deg(P_0) = 1024 \cdot 2 = 2048$. We need $\deg(P) \geq 2024$. Our value 2048 satisfies this condition. (If we needed an even higher degree, we could just choose a larger $k$. For instance, $k = 11$ would give $\deg(P) = 4096$.)

4. **The functional equation**: As shown in Step 2, the choice $P(x) = R(P_0(x))$ and $Q(x) = Q_0(x)$ reduces the equation $(*)$ to $R(Q_0(P_0(x))) = Q_0(R(P_0(x)))$. Our construction of $R(x)$ ensures it commutes with $Q_0(x)$, so this identity holds for all inputs, and thus for all real numbers $x$.

We have successfully constructed polynomials $P(x)$ and $Q(x)$ that satisfy all the required conditions. For completeness, the polynomials are: - $Q(x) = \frac{1}{4}x^2 + 2x + 3$. - $P(x) = S_{10}(\frac{1}{4}x^2 + x + 4) - 4$, where $S(y) = \frac{1}{4}y^2 - 1$ and $S_{10}$ is the 10th iterate of $S(y)$.

This completes the proof that such polynomials exist.

---

**Case 4: review from GPT-5-mini**

`<verification>false</verification>` The fatal error is in Step 2: the author claims $Q_0(x) = L^{-1}(S(L(x)))$ with $L(x) = x + 4$ and then uses this to conclude that $R(x) = L^{-1}(S_k(L(x)))$ commutes with $Q_0$ whenever $S_k$ commutes with S. But $Q_0(x) = 1/4(x + 4)^2 - 1$ equals $S(L(x))$, not $L^{-1}(S(L(x)))$. That mistaken conjugation breaks the commuting argument: the equality $R(Q_0(x)) = Q_0(R(x))$ does not follow from $S_k \circ S = S \circ S_k$ under the provided definitions, and the displayed $R(x) = S_k(x+4) - 4$ need not commute with $Q_0$. (Thus the key commuting-polynomial construction is invalid as written.)

---

The manually annotated label in the dataset is true (7 points), while our method found a typo in it:

---

**Case 4: our comment**

The review fromi GPT-5-mini actually points out a typo that $Q_0(x) \neq L^{-1}(S(L(x)))$. However, the choice of $S$ does not really matter so we can fix the proof easily.

---

**Case 5: problem**

Prove that there exist two real-coefficient polynomials $P(x)$ and $Q(x)$ satisfying the following condition:
(Condition) The degree of the polynomial $P(x)$ is at least 2024, the degree of $Q(x)$ is at least 2, and for any real number $x$, the following holds:

$$P(Q(x) - x - 1) = Q(P(x))$$

---

**Case 5: proof**

Let $n = 2024$. We are looking for real-coefficient polynomials $P(x)$ and $Q(x)$ such that $\deg(P) \geq n$ and $P(Q(x) - x - 1) = Q(P(x))$ for all real $x$.

Let $\deg(P) = d_P$ and $\deg(Q) = d_Q$. Comparing the degrees of both sides of the equation $P(Q(x) - x - 1) = Q(P(x))$, we have $\deg(Q(P(x))) = d_Q d_P$. $\deg(P(Q(x) - x - 1)) = d_P \deg(Q(x) - x - 1)$. If $d_Q > 1$, $\deg(Q(x) - x - 1) = d_Q$, so $\deg(P(Q(x) - x - 1)) = d_P d_Q$. If $d_Q = 1$, let $Q(x) = cx + d$ with $c$ e0. Then $Q(x) - x - 1 = (c-1)x + d - 1$. If $c$ e1, $\deg(Q(x) - x - 1) = 1 = d_Q$, so $\deg(P(Q(x) - x - 1)) = d_P \cdot 1 = d_P$. The degree of $Q(P(x))$ is $1 \cdot d_P = d_P$. If $c = 1$, $Q(x) = x + d$. $Q(x) - x - 1 = d - 1$. $\deg(P(Q(x) - x - 1)) = \deg(P(d-1)) = 0$. The degree of $Q(P(x)) = P(x) + d$ is $d_P$. For $0 = d_P$, $P(x)$ must be a constant. Let $P(x) = k$. Then $P(Q(x) - x - 1) = k$ and $Q(P(x)) = Q(k) = k + d$. So $k = k + d \implies d = 0$. $Q(x) = x$. In this case $P(Q(x) - x - 1) = P(-1)$ and $Q(P(x)) = P(x)$. So $P(x)$ must be a constant $k$ such that $P(-1) = k$. This works, but we need $d_P \geq 2024$. So we must have $d_Q \geq 1$ and $c$ e1 if $d_Q = 1$.

Let's try to find a solution with $d_Q = 1$. Let $Q(x) = cx + d$ with $c$ e1. The condition is $P((c-1)x + d - 1) = cP(x) + d$. Let $d_P = n = 2024$. Let $P(x) = a_n x^n + \cdots + a_0$ with $a_n$ e0. Comparing the leading coefficients, $a_n((c-1)x)^n + \cdots = c(a_n x^n) + \dots$. $a_n(c-1)^n x^n + \cdots = ca_n x^n + \dots$. Since $a_n$ e0, we must have $(c-1)^n = c$. Let $n = 2024$. We need to show that there exists a real number $c$ e1 such that $(c-1)^{2024} = c$. Consider the function $f(t) = (t-1)^{2024} - t$ for $t \in \mathbb{R}$. $f(t)$ is continuous. $f(0) = (0-1)^{2024} - 0 = 1 > 0$. $f(1) = (1-1)^{2024} - 1 = -1 < 0$. By the Intermediate Value Theorem, there exists $c_1 \in (0, 1)$ such that $f(c_1) = 0$. Since $c_1 \in (0, 1)$, $c_1$ e1. $f(3) = (3-1)^{2024} - 3 = 2^{2024} - 3 > 0$. By the Intermediate Value Theorem, there exists $c_2 \in (1, 3)$ such that $f(c_2) = 0$. Since $c_2 \in (1, 3)$, $c_2$ e1. Let $c$ be any such real root. For example, $c = c_1$ or $c = c_2$. Now we need to find a polynomial $P(x)$ of degree $n = 2024$ satisfying $P((c-1)x + d - 1) = cP(x) + d$ for some $d$. Let's try $d = 1$. We need $P((c-1)x) = cP(x) + 1$. Let $P(x) = ax^n + b$. $a((c-1)x)^n + b = c(ax^n + b) + 1$. $a(c-1)^n x^n + b = acx^n + cb + 1$. $acx^n + b = acx^n + cb + 1$. This requires $b = cb + 1$, so $b(1 - c) = 1$. Since $c$ e1, we can choose $b = \frac{1}{1-c}$. Let $a = 1$. Then $P(x) = x^n + \frac{1}{1-c}$ is a polynomial of degree $n = 2024$. Let $P(x) = x^{2024} + \frac{1}{1-c}$ and $Q(x) = cx + 1$. $P(x)$ has degree $2024 \geq 2024$. $P(x)$ and $Q(x)$ have real coefficients since $c$ is real. Let's verify the condition: $P(Q(x) - x - 1) = P(cx + 1 - x - 1) = P((c-1)x) = ((c-1)x)^{2024} + \frac{1}{1-c} = (c-1)^{2024}x^{2024} + \frac{1}{1-c} = cx^{2024} + \frac{1}{1-c}$. $Q(P(x)) = cP(x) + 1 = c(x^{2024} + \frac{1}{1-c}) + 1 = cx^{2024} + \frac{c}{1-c} + 1 = cx^{2024} + \frac{c+1-c}{1-c} = cx^{2024} + \frac{1}{1-c}$. The condition $P(Q(x) - x - 1) = Q(P(x))$ is satisfied. Thus, such polynomials $P(x)$ and $Q(x)$ exist.

Final Answer: The final answer is $\boxed{P(x) \text{ and } Q(x) \text{ exist}}$

---

**Case 5: review from qwen3-30b-a3b**

`<verification>false</verification>` The solution incorrectly constructs $Q(x)$ as a linear polynomial (degree 1), violating the problem's requirement that $\deg(Q) \geq 2$. This critical error undermines the correctness of the solution.

The manually annotated label in the dataset is true (7 points), while our method found a critical error in it:

---

**Case 5: our comment**

$\deg(Q)$ has to be at least 2 so this construction cannot be accepted.

---

**Case 6: problem**

There are 120 students in DeepMind highschool, and each student 'loves' some of the students. Here 'love' is not always a mutual relationship; a student A may love B while B may not love A, and it is also possible that a student loves oneself. For the set $F$ of all $2^{120}$ subsets of the students, define the function $f : F \to F$ as the function that maps each $X \in F$ to the set of students loved by one or more students in $X$. For $A, B \in F$, $A$ and $B$ are said to have a 'lovely relationship' iff there exists a natural number $k$ such that $f^k(A) = B$. Now we want to select distinct elements $A_1, A_2, \cdots A_t$ from $F$ such that for any $1 \leq i < j \leq t$, $A_i$ and $A_j$ have a lovely relationship. Let $M(f)$ be the maximum possible value of $t$. Show that $M(f) \leq 2^{70}$. ", "proof": "Let $S$ be the set of 120 students, and $F = 2^S$ be the power set of $S$. The function $f : F \to F$ is defined as $f(X) = \bigcup_{u \in X} N(u)$, where $N(u)$ is the set of students loved by student $u$. This defines a directed graph $G = (S, E)$ where $(u, v) \in E$ if $u$ loves $v$.

---

**Case 6: proof**

The function $f$ is a linear map over the vector space $\mathbb{F}_2^{120}$, where subsets are identified with their characteristic vectors. The union operation corresponds to vector addition.

A 'lovely relationship' between two sets $A, B \in F$ means that one can be reached from the other by repeated application of $f$. A collection of distinct sets $\{A_1, \ldots, A_t\}$ where any pair has a lovely relationship must form a chain under this reachability relation. We can reorder these sets to form a simple path in the functional graph of $f$. The maximum possible value of $t$, denoted $M(f)$, is the length of the longest such path. A maximal path can be written as $A_0, A_1, \ldots, A_{t-1}$, where $A_{i+1} = f(A_i)$ for $i = 0, \ldots, t-2$, and all $A_i$ are distinct.

Our proof strategy is to analyze the structure of the graph $G$ by decomposing it into a "core" part and an "acyclic" part. We will use the properties of the iterates of $f$ on the whole set $S$.

Let $L_0 = S$, and define a sequence of subsets $L_k = f^k(S)$ for $k \geq 1$. The function $f$ is monotone, i.e., if $X \subseteq Y$, then $f(X) \subseteq f(Y)$. Since $L_1 = f(S) \subseteq S = L_0$, monotonicity implies $f(L_1) \subseteq f(L_0)$, so $L_2 \subseteq L_1$. This gives a descending chain of subsets: $S = L_0 \supseteq L_1 \supseteq L_2 \supseteq \cdots$.

Since $S$ is finite, this sequence must stabilize. Let $p$ be the first integer such that $L_p = L_{p+1}$. Let $C_f = L_p$. This is the "image core" of the graph under $f$. From $L_{p+1} = f(L_p)$, the stability condition $L_p = L_{p+1}$ means $f(C_f) = C_f$. This implies that the function $f$, when restricted to the set $C_f$, acts as a permutation on the elements of $C_f$. Let's call this permutation $\pi$. Let $x \in C_f$. Since $f(C_f) = C_f$, every element in $C_f$ must have at least one preimage in $C_f$. Since $C_f$ is finite, this means $f$ restricted to the elements of $C_f$ is a bijection, i.e., a permutation $\pi$. This implies that for each $x \in C_f$, its neighborhood $N(x)$ must be a singleton set $\{\pi(x)\}$, and $\pi(x) \in C_f$.

Now we analyze the structure of any trajectory $A_0, A_1, \ldots, A_{t-1}$. Let $m$ be the first index in the sequence such that $A_m$ is a subset of $C_f$. So $A_i \not\subseteq C_f$ for $i < m$, and $A_m \subseteq C_f$.

The property $f(C_f) = C_f$ implies that for any $u \in C_f$, $N(u) \subseteq C_f$. Let $X \subseteq C_f$. Then $f(X) = \bigcup_{u \in X} N(u) \subseteq C_f$. This means that once a set in the trajectory is a subset of $C_f$, all subsequent sets in the trajectory are also subsets of $C_f$. So $A_i \subseteq C_f$ for all $i \geq m$. The total length of the trajectory is $t = m + (t - m)$. The first part of the trajectory has length $m$, and the second part has length $t - m$. The two parts are disjoint because sets in the first part are not

subsets of $C_f$, while sets in the second part are.

Let's bound the length of the pre-core part of the trajectory, $m$. Let $U = S \setminus C_f$. For $i < m$, $A_i \cap U \, e\emptyset$. Let $A_i^U = A_i \cap U$ and $A_i^C = A_i \cap C_f$. $A_{i+1} = f(A_i) = f(A_i^C \cup A_i^U) = f(A_i^C) \cup f(A_i^U)$. Since $N(C_f) \subseteq C_f$, we have $f(A_i^C) \subseteq C_f$. Therefore, the part of $A_{i+1}$ outside the core is determined solely by the part of $A_i$ outside the core: $A_{i+1}^U = A_{i+1} \cap U = (f(A_i^C) \cup f(A_i^U)) \cap U = f(A_i^U) \cap U$. The sequence of non-core parts $(A_i^U)_{i \geq 0}$ evolves autonomously.

The set $C_f = L_p = \bigcap_{k \geq 0} L_k$ consists of all vertices that can be reached by paths of arbitrary length from $S$. This means any vertex not in $C_f$ cannot reach a cycle. The subgraph $G[U]$ induced by $U = S \setminus C_f$ must be a Directed Acyclic Graph (DAG).

Let $v_1, v_2, \ldots, v_{|U|}$ be a topological sort of the vertices in $U$. For any edge $(v_i, v_j)$ in $G[U]$, we have $i < j$. For any non-empty subset $Y \subseteq U$, define its rank as $\min_r(Y) = \min\{i \mid v_i \in Y\}$. Let $Y_i = A_i^U$. Suppose $Y_i \, e\emptyset$. Let $j = \min_r(Y_i)$. $Y_{i+1} = f(Y_i) \cap U = \bigcup_{u \in Y_i}(N(u) \cap U)$. Any vertex $w \in N(v_j) \cap U$ must have an index greater than $j$. The same applies to any other $u \in Y_i$ with index greater than $j$. Thus, for any vertex in $Y_{i+1}$, its index in the topological sort is greater than $j$. This means $\min_r(Y_{i+1}) > \min_r(Y_i)$. The sequence of ranks $\min_r(A_0^U), \min_r(A_1^U), \ldots$ is strictly increasing as long as the sets are non-empty. Since the rank is at most $|U|$, the sequence of non-empty sets $(A_i^U)$ can have length at most $|U|$. $A_{m-1}^U \, e\emptyset$ and $A_m^U = \emptyset$. So $m \leq |U|$. Let $k = |C_f|$. Then $|U| = 120 - k$. We have $m \leq 120 - k$.

Now let's bound the length of the core part of the trajectory, $t - m$. For $i \geq m$, the sets $A_i$ are subsets of $C_f$. As we established, for any $x \in C_f$, $N(x) = \{\pi(x)\}$ where $\pi$ is a permutation of $C_f$. So, for any $X \subseteq C_f$, $f(X) = \bigcup_{x \in X}\{\pi(x)\} = \pi(X)$. The trajectory for $i \geq m$ is given by $A_{i+1} = \pi(A_i)$. The sequence is $A_m, \pi(A_m), \pi^2(A_m), \ldots, A_{t-1}$. Since these sets must be distinct, the length $t - m$ is bounded by the order of the permutation $\pi$ in the symmetric group on $2^{C_f}$. This is the order of the element $A_m$. The maximum length is the maximum cycle length in the permutation group induced by $\pi$ on $2^{C_f}$. This is bounded by the order of $\pi$ acting on $C_f$. The order of $\pi \in S_k$ is the least common multiple (lcm) of the lengths of its disjoint cycles. The maximum possible order for a permutation of $k$ elements is given by Landau's function $g(k)$, defined as $g(k) = \max_{n_1 + \cdots + n_r = k} \text{lcm}(n_1, \ldots, n_r)$. So, $t - m \leq g(k)$.

Combining the bounds for both parts, we get: $t = m + (t - m) \leq (120 - k) + g(k)$.

We need to find an upper bound for $h(k) = 120 - k + g(k)$ for $k \in \{0, 1, \ldots, 120\}$. Landau's function $g(k)$ grows subexponentially. The asymptotic growth is $\ln g(k) \sim \sqrt{k \ln k}$. Let's check some values: $h(0) = 120 - 0 + g(0) = 120 + 1 = 121$ (a trajectory on $\emptyset$ has length 1). $h(1) = 120 - 1 + g(1) = 119 + 1 = 120$. $h(10) = 110 + g(10) = 110 + \text{lcm}(2, 3, 5) = 110 + 30 = 140$. $h(20) = 100 + g(20) = 100 + \text{lcm}(3, 4, 5, 7) = 100 + 420 = 520$. $h(40) = 80 + g(40) = 80 + \text{lcm}(3, 5, 7, 8, 11) = 80 + 9240 = 9320$. $h(70) = 50 + g(70)$. $g(70) = \text{lcm}(3, 4, 5, 7, 11, 13, 17) = 1161160$. $h(70) = 1161210$. For $k = 120$, $h(120) = g(120)$. A partition of 120 into prime powers that gives a large lcm is $3, 4, 5, 7, 11, 13, 17, 19, 23$. The sum is 102. Remaining 18. Add 1. $g(102)$ is huge. For $k = 120$, $g(120)$ is known to be $\approx 3.2 \times 10^{10}$.

In all cases, the value of $h(k) = 120 - k + g(k)$ is significantly smaller than $2^{70}$. $2^{10} \approx 10^3$. $2^{70} = (2^{10})^7 \approx (10^3)^7 = 10^{21}$. The maximum value of $h(k)$ for $k \in [0, 120]$ is many orders of magnitude smaller than $2^{70}$. Therefore, we have shown that for any graph $G$, $M(f) \leq \max_{0 \leq k \leq 120}(120 - k + g(k)) \ll 2^{70}$.

This shows that $M(f) \leq 2^{70}$.

---

**Case 6: review from qwen3-30b-a3b**

`<verification>false</verification>`
The solution incorrectly claims that the function $f$ corresponds to vector addition in $\mathbb{F}_2^{120}$, which is invalid because $f$ is defined via union (not symmetric difference). It also wrongly asserts that $f$ restricted to the core $C_f$ is a permutation, which fails to account for possible non-injective behavior of $f$. Additionally, the analysis of the core assumes each $N(x)$ is a singleton, which is not guaranteed by the problem's conditions. These critical errors undermine the proof's validity."

The case shows that model may misunderstand definitions if the explanation is unclear in some sense. Actually, the human expert misunderstands it as well at first.

**Case 6: our comment**

The proof is true. The model misunderstands the definition of the function $f$. However, It can easily be misunderstood due to the unclear explanation.

