# OpenReview forum: "Pessimistic Verification for Open-Ended Math Questions"
_ICML.cc/2026/Conference — ICML 2026 regular_

### Official Review · Reviewer_o8pG · 2026-02-21

**Soundness:** 3
**Presentation:** 4
**Significance:** 3
**Originality:** 2
**Overall Recommendation:** 4
**Confidence:** 2

**Summary:**

The paper suggests a new framework called "pessimistic verification": running a few parallel verifiers and reject if any of them rejects (finds an error). The goal is to increase the verification ability efficiently compared to other methods like CoT.
The authors argue that the main bottleneck is finding errors and not confirmation. They emphasize the error-centric verification philosophy, meaning that even a single error is enough to reject a proof.
They propose three variants of the pessimistic verification framework and examine their performance: how they improve error detection, how they scale and how they improve the balance between the false positive and false negative rates.

**Compliance With Llm Reviewing Policy:**

Affirmed.

**Final Justification:**

The rebuttal clarified the intuition behind the cost/accuracy tradeoff, addressed the limitations of QiuZhen-Bench more directly, and gave a reasonable discussion of how the approach may extend beyond math. The main remaining weakness, in my view, is limited conceptual novelty: the core idea is simple and the paper's contribution is best understood as primarily empirical, namely demonstrating that pessimistic verification and its progressive variant work well in practice. I would still encourage the authors to add some theoretical analysis in the revision, but overall the rebuttal addressed my main questions and did not change my positive assessment.

**Key Questions For Authors:**

1) Can you formally analyze of the cost-accuracy tradeoff of pessimistic verification? (where accuracy here means error detection), and the cost can be e.g. the number of verification rounds)

2) How can you ensure that evaluation on  QiuZhen-Bench does not partially reflect alignment with the annotator?

3) Do you expect your approach to generalize beyond the math domain, e.g. for coding? How do you believe it will perform there? What are the challenges?

**Limitations:**

yes

**Strengths And Weaknesses:**

Strengths

(-) The problem is clear and well-motivated, and the idea is very simple and practical.

(-) The paper is well-written. The related work survey seems thorough. The presentation in general is very appealing and the figures are helpful.

(-) The recursive technique in the prog@n/k variant is interesting and its good performance is even counter-intuitive to me (as these approaches tend to be inefficient, even with early pruning of wrong proofs).

(-) The observation that the existing verification benchmarks are unreliable for frontier models is important.

Weaknesses

(-) The theoretical idea does not seem novel. The idea of repetition to reduce error is classical (e.g., in Interactive Proofs it is one of the most basic tools), so the conceptual novelty here is limited. I also assume it is already widely used in practice, at least implicitly. This doesn't mean that the paper's contribution on the empirical level is significant, only that the conceptual contribution is rather straightforward.
The same holds with respect to the two underlying philosophies in section 5.

(-) This means that the contribution is mostly empirical, as there are no theoretical-mathematical guarantees like a cost-accuracy tradeoff.
Hence, I think that statements like "error detecting is the key ability that separates strong verifiers" is not theoretically established.

(-) QiuZhen-Bench in annotated by GPT5, and this may introduce some circularity: similar frontier models may partially align with the annotator model, and the evaluation will not be independent verification ability.

---

> ### Author Rebuttal · Authors · 2026-03-27
>
> Thank you very much for your detailed review on this paper, here I will provide more explanations on the weaknesses and questions mentioned in the review. I hope this can resolve some of your concerns!
>
> **Q1 & W1, W2**
>
> Yes, and that is really a good question. We can begin by considering the error detection process. Let $p$ be the probability that an LLM can reveal the error in the given false proof. Pessimistic verifications in this paper adopts two key methods:
>
> 1. **Vertical split** and piecewise verification, which can increase the performance of error detection when the reviewer only looked through the problem once (while requiring $n$ reviews per iteration). So we can denote this probability by $p_{v@n}$, while $p_{v@1} = p$.
> 2. **Pessimistic voting** which marks the proof as false if any one of the parallel reviews reports false.
>
> Let the number of vertical splits in each review iteration be $\{n_i\}_{i=1}^I$, the error detection probability and number of API calls can of pessimistic verification families can be written as
>
> $$
> p_\text{pess} = 1 - \prod_{i=1}^I (1 - p_{v@n_i}), \quad \text{API calls} = \sum_{i=1}^I n_i
> $$
>
> In simple pessimistic verification we set $n_i = 1, \forall i$, in vertical verification we set $I = 1$ (only one parallel iteration), and in progressive verification we set $n_i = 2^{i-1}$. It is clear that $p_\text{pess}$ is constantly higher than each $p_{v@n}$. Then the problem becomes, with a fixed computational budget, how can we arrange $\{n_i\}_{i=1}^I$ to obtain the best error detection performance? Simple and vertical pessimistic verifications can represent two extreme case, with the most or only one iterations $I$. Empirically both of them tend to be less performant than progressive verification, although it might still not be the most performant one.
>
> We can also do similar analysis on the positive samples and majority voting. This also highlights the two philosophies introduced in this paper:
>
> 1. Simple parallel reviews can help find more errors and help improve the solution quality in an agentic workflow, but their stability to maintain positive samples is not well studied. We primarily used Balanced F1 score to measure the performance, and showed that under this scheme the overall performance also steadily increases.
> 2. Using vertical split and require for detailed analysis over multiple simple verification is a new technique, and progressive scheme also exhibits \~2x higher efficiency than simple pessimistic verification.
>
> This is only a minimal theoretical explanation with many details excluded to make this rebuttal clean. We are also willing to include a more detailed theoretical analysis of these algorithms in the appendix of the camera-ready version of this paper.
>
> **Q2 & W3**
>
> QiuZhen-Bench is a auxiliary benchmark on the advanced-level math, while most manually labeled verification benchmarks still focus on elementary math. We only evaluated weaker models on this benchmark with the annotation of GPT-5, and the results showed a similar trend as other verification benchmarks or problem-solving benchmarks, so these results are reliable to some extent.
>
> However, since distillation commonly exists in the training process of LLMs, we can not completely distinguish the actual performance and the alignment with the annotator. As the performance of frontier LLMs continues improving, it is becoming increasingly challenging to construct accurate benchmarks on them. Even IMO-GradingBench, the benchmark created by many experts in Google, still contains a lot of annotation errors, and the performance of frontier LLMs rapidly saturated. We have added explicit notations on the results of QiuZhen-Bench in the paper, in order not to mislead the readers. This unreliability may not harm the core results in our work.
>
> **Q3**
>
> Yes, and I believe this scheme will be increasingly important for industrial-level coding tasks or research-level math, which are both long-form tasks that requires long action trajectories, reasoning process and complex background information.
>
> Currently popular coding agents already adopts a similar workflow in the debugging process, say: writing code -> designing / running tests -> review existing codebase -> revise buggy code. In this process both philosophies can help a lot:
>
> 1. It can use an error-centric parallel reviews to find more subtle errors in code, which might be common and especially harmful in complex algorithms.
> 2. The agent should look into the details of each line of code when reviewing, instead of briefly go through the overall code structure multiple times.
>
> We can also think of some challenges for coding tasks, which requires more explorations in the following works.
>
> 1. How to integrate coding tests and this pure reasoning-based verifications?
> 2. How to heuristicly search for related code before applying detailed verification? (This might also be some extended version of "progressive" verification)

---

> > ### Author Rebuttal · Reviewer_o8pG · 2026-04-02
> >
> > Thank you for the helpful rebuttal. My concerns are fully resolved.
> > I appreciate the added clarification on the intuition behind the cost/accuracy tradeoff, the discussion of the limits of QiuZhen-Bench and the comments on how the approach may extend beyond math. The rebuttal also helps clarify that the paper's main contribution is primarily empirical, which I think is indeed the right framing for it. I encourage you nonetheless to include some theoretical analysis of these algorithms in the revision.
> > Overall, the response addressed my main questions and improved my understanding of the paper.

---

> > > ### Author Response · Authors · 2026-04-03
> > >
> > > Thank you again for you approval and constructive review! We also believe that theoretical analysis will further enhance the validity of this work. So... would you like to raise your score in the review? If you have more questions or concerns, we are also willing to continue to communicate at any time！

---

### Official Review · Reviewer_yqXc · 2026-03-09

**Soundness:** 2
**Presentation:** 2
**Significance:** 3
**Originality:** 3
**Overall Recommendation:** 4
**Confidence:** 3

**Summary:**

This paper introduces a novel approach to verifying solutions to open-ended mathematical problems using large language models (LLMs). The authors propose a pessimistic verification method, where the solution is rejected if any of several verifiers identify an error. They also introduce a progressive version of pessimistic verification, which further divides the proof into smaller chunks for more efficient error detection. The paper compares the proposed methods with existing verification techniques such as standard verification and majority voting across various benchmarks, including the IMO 2025, MathArena Apex 2025,and IMO-GradingBench datasets. The results demonstrate that progressive pessimistic verification not only improves the accuracy of error detection but also enhances the efficiency of the process.

**Compliance With Llm Reviewing Policy:**

Affirmed.

**Final Justification:**

Thank you for the rebuttal. The authors provided useful clarifications on several points, and the response improves the transparency of the paper. In particular, the additional discussion around evaluation choices, runtime/cost reporting, and parts of the methodology helps make the work easier to understand.

That said, I still believe the paper has some non-trivial weaknesses. Several of my original concerns remain only partially addressed, especially regarding the evaluation design, the strength of the empirical support for some claims, and the extent to which the conclusions are justified by the current evidence. While the rebuttal adds helpful context, it does not fully resolve these issues.

Overall, my assessment remains broadly unchanged. I see merit in the paper: the problem is interesting, the approach has some novelty, and the work may be of interest to the community. At the same time, I do not find the evidence strong enough to remove my reservations entirely. Therefore, my final position is essentially neutral/borderline: I can see reasonable arguments for acceptance, but I also think rejection would be defensible given the remaining weaknesses. My original leaning remains appropriate.

**Key Questions For Authors:**

1. Clarify Hyperparameter Selection: A more detailed explanation of how hyperparameters were chosen and how they impact the results could be helpful for future researchers attempting to replicate or extend the work.

2. Extend Experiments to Larger Datasets: While the paper focuses on smaller models and datasets, conducting similar experiments on larger models, particularly those involving high-dimensional or long-form proofs, would solidify the method's scalability.

3. Expand Formal Verification Comparison:
A deeper comparison with formal verification methods would provide a more comprehensive understanding of where the proposed method stands relative to existing, more established verification techniques

**Limitations:**

yes

**Strengths And Weaknesses:**

Strengths

1. Novel Approach: The concept of pessimistic verification, particularly the progressive version, is a strong contribution to the verification of mathematical proofs using LLMs. It addresses the problem of error detection with multiple rounds of verification and shows promise in improving both performance and efficiency.
2. Efficiency Focus: The paper not only focues on verification accuracy but also emphasizes computational efficiency, an important aspect in real-world applications. The proposed method manages to reduce the token usage, which is a key consideration in large-scale LLM applications.
3. Reproducibility: The experiments are well-detailed, and the datasets used  are relevant and rigorous. The authors have shared the case studies and datasets, making their research reproducible and valuable for future work.

Weaknesses

1. Hyperparameter Tuning: While the authors mention that their experiments involve different hyperparameter settings (e.g., progressive@3/6), the methodology for selecting these hyperparameters is not deeply explained. It would be helpful to understand how different settings affect the tradeoff between accuracy and efficiency.Comparison with Formal
2. Minor Clarity Issues in Methodology: Some sections could benefit from a more thorough explanation of the technical details. For example, the distinction between vertical and progressive pessimistic verification could be clearer, especially for readers unfamiliar with these methods. Visual aids, like diagrams for method comparisons or more detailed flow charts could enhance clarity.

---

> ### Author Rebuttal · Authors · 2026-03-27
>
> Thank you for your review! Here are the responses to your concerns on this paper:
>
> **Q1 & W1, W2: Clarify Hyperparameter Selection**
>
> In fact we have included the experimental results under different hyperparameter settings, see the efficiency experiments in *Figure 4*. These data points are evaluated under different hyperparameter settings. For example in this graph the data points of the blue line (progressive pessimistic verification) can separately reflect the cost and performance under "progressive@1/6", "2/6", "3/6", and "4/6", from left to right. The second parameter in progressive verification, the minimal chunk length, is auxiliary and only has little impact on the performance with proper values. At the same time the red line is simple pessimistic verification with different number of reviewers.
>
> So we choose "progressive@3/6" and "pessimistic@12" for most experiments primarily according to this principle: These parameters should be large enough to reflect the charactistic of different methods, while the experimental costs can be controlled in a reasonable amount.
>
> We then used progressive verification with 4 iterations for the following problem-solving tasks, since these datasets only contain a small number of problems and this will not introduce too much token usage. We will clarify this hyperparameter selection strategy and the technical details in the camera-ready version of this paper.
>
> **Q2: Extend Experiments to Larger Models / Datasets**
>
> We also agree that further extend the experiments will further solidify the methods' scalability. Currently our experiments already covers a wide range of models and datasets. In *Figure 5* we thoroughly compared the performance from the smallest "Qwen3-1.7B" to "GPT-5" on three verification benchmarks, and validated the performance gains. In the latter problem-solving benchmarks with reviewer assistance, we also confirmed around \~2x higher efficiency and better performance on problems with IMO-level difficulty and beyond.
>
> However, there are still notable difficulties to further extend our experiments. There is no well-known benchmarks on long-form research-level math with higher difficulty and complexity than IMO, and manually reviewing these results would require a large amount of labour from experts. There are some works exploring research-level math, such as Alethia from Google [1]. It also contains a natural language based verifier as one of its core components, while its technical details were not explained. We will also continue working on this direction, which certainly requires to further validating the effectiveness of our verifier on these frontier problems, but these experimental results are the best we can do currently.
>
> **Q3: Expand Formal Verification Comparison**
>
> In Section 4.2 we have included a brief discussion on formal verification, and here we can provide more comparison on these methods and verifications in natural language (which we call informal verification).
>
> Formal verifiers are fully rigorous, symbolic methods for verifying math proofs. Typically it also requires an specially designed agentic workflow to efficiently conduct verification on a given proof. The **positive results** it reports is almost completely reliable, and recently there are also some works claiming the automatic verification of results with Fields medal [2] (It is an easiest special case though).
>
> However, the **negative results** in formal verification tend to be less informative. It can not distinguish errors in the original proof or in the autoformalization process, indicating that formal verification can not actually reject a proof. This is an intresting phenomenon that is contrary to informal verifications, which we indicate that negative reviews are the most valuable while positive ones are not. Most formal workflows also integrates informal process for problem solving and proof revising, but they are constantly less performant compared to pure informal workflows in problem-solving tasks. Most works exploring automatic problem-solving in frontier math choose to use informal verification in their workflows [1][3].
>
> One common property of these verification methods is that they both consumes a large amount of resources to perform. Both of them may require millions of tokens to verify and accept a contest-level problem, so there is an urgent need to find a more efficient method for verification. Our progressive variant showed remarkable increment over simple parallel verification, but this problem is still far from being solved.
>
> [1] Alethia. https://arxiv.org/abs/2602.10177
>
> [2] Formalising Sphere Packing in Lean. https://thefundamentaltheor3m.github.io/Sphere-Packing-Lean/
>
> [3] AI Mathematician. http://arxiv.org/abs/2505.22451

---

> > ### Author Rebuttal · Reviewer_yqXc · 2026-04-03
> >
> > I think your points are valid, and I will keep my score unchanged.

---

### Official Review · Reviewer_BdY7 · 2026-03-12

**Soundness:** 3
**Presentation:** 2
**Significance:** 2
**Originality:** 2
**Overall Recommendation:** 3
**Confidence:** 5

**Summary:**

This paper studies LLM-based verification of open-ended mathematical proofs and argues that the main bottleneck is error detection rather than recognizing correct proofs. It proposes three simple test-time workflows--simple, vertical, and progressive pessimistic verification--that reject a proof if any verification pass identifies an error. The idea is intuitive and the empirical trend is interesting: pessimistic aggregation often improves TNR and balanced F1 over single-pass verification and majority voting

**Compliance With Llm Reviewing Policy:**

Affirmed.

**Key Questions For Authors:**

1. How were key evaluation choices selected, especially progressive@3/6 and the 300-example subsets? Was there any held-out validation or tuning protocol independent of the reported benchmarks?

2. Can the authors compare against the strongest verifier/grading workflows already cited in the paper, rather than only single-pass verification, majority voting, and long-CoT baselines?

3. Can the efficiency claim be supported with actual runtime/API cost measurements, and can the annotation-error claim be strengthened with a larger independent adjudication study?

**Limitations:**

The limitation is not disscussed.

**Strengths And Weaknesses:**

# Strengths
The paper isolates a plausible and important failure mode in proof verification, namely missed errors. The proposed workflows are simple, easy to understand, and potentially useful in practice. I also found the comparison against majority voting informative: it helps support the claim that the aggregation rule matters, not just repeated sampling.

# Weaknesses

1. The contribution is methodologically modest: the simple variant is essentially an OR rule over repeated verifier samples, the vertical variant is chunk-wise checking, and the progressive variant is a hierarchical combination of these ideas. That can still be publishable, but then the empirical validation needs to be especially strong.

2. Experimental support is not strong enough for the scope of the claims. The baseline set is limited (mostly single-pass, majority voting, and long-CoT), despite the paper citing stronger neighboring verifier/grading workflows. In addition, the evaluation uses subsets of larger datasets without a clear sampling/tuning protocol, and reports no uncertainty estimates, variance, or significance despite stochastic decoding.

3. Some headline claims are overstated relative to the evidence.
In particular, the claims about efficiency and annotation error are not yet sufficiently supported. The efficiency analysis relies mainly on a token-based proxy rather than actual runtime or cost measurements. The annotation-error narrative is based on a small post hoc analysis and does not yet justify broader claims about benchmark unreliability or near-human-expert performance.

4. QiuZhen-Bench is not strong independent evidence.
Since this benchmark is labeled with GPT-5 using the authors’ own preferred verification workflow, it is not ideal for supporting the paper’s strongest empirical claims. At best, it is supplementary evidence.

---

> ### Author Rebuttal · Authors · 2026-03-30
>
> Thank you for your careful review of our work! Here I will provide more explanations on the questions.
>
> **Q1: Key evaluation choices**
>
> Reviewer yqXc also asked about parameter selection. We mainly use progressive@3/6 and pessimistic@12 because these settings are large enough to reflect the characteristics of each method, while keeping evaluation cost manageable.
>
> In the efficiency results (Figure 4), the plotted points are obtained under multiple hyperparameter settings. Performance gains are consistently observed across many settings, and the progressive variant is almost always the strongest.
>
> The 300-example subset consists simply of the first 300 examples in the original IMO-GradingBench ordering, chosen only for cost control. For Hard2Verify, we evaluate on the full dataset. We confirm that no held-out validation set or tuning protocol was used in our experiments.
>
> **Q2: Comparison with stronger baselines**
>
> We cited a range of related verifier / grading workflows, but did not include them as baselines for two main reasons: They target different objectives, so the results are not directly comparable.
>
> Many of them implicitly use a form of simple pessimistic verification, but do not study it explicitly as a verification method. The first category includes many grading workflows discussed in Related Work. These methods mainly aim to align workflow scores with human judgments, which involves substantial subjectivity and does not match our benchmark target. They are also not designed to assist agent problem-solving.
>
> In fact, in benchmark papers such as Hard2Verify [1] and OPC [2], the main compared verification baseline is single-pass verification with long CoT. Hard2Verify [1] even states in Sec. 5.1 that “sequential scaling brings substantive gains, whereas parallel scaling does not.” Their parallel variants (e.g., majority voting / best-of-N) did not improve review performance. In contrast, our results show that pessimistic verification can yield clear gains. Notably, these benchmarks already report near-human-expert verification performance.
> The second category includes DeepSeekMath-V2 [3] and related works [4][5]. These methods naturally use multiple parallel reviews to identify issues and refine solutions, but they are not evaluated as verification methods. Their goal is to improve problem-solving by surfacing as many potential errors as possible, while its TPR and balanced F1 score are not considered.
>
> This distinction is one of our main contributions: we show that simple pessimistic verification is also **stable as a verification method**, consistently improving balanced F1. Moreover, if treated as the commonly adopted baseline discussed in Sec. 2.1, our proposed progressive pessimistic verification is still at least 2× more efficient on both verification and problem-solving benchmarks.
>
> **Q3: Efficiency measurements and annotation error claims**
>
> Yes, and here are some auxiliary results on the performance and per sample run time and API costs.
>
> |Model|Method|Bal. F1|eq. tokens|run time / seconds|cost / \$|
> |-|-|-|-|-|-|
> |gpt-5-mini|standard|0.78|2,975.5|6.15|0.0060|
> |gpt-5-mini|pes@12|0.87|35,675.8|78.55|0.0714|
> |gpt-5-mini|prog@3/6|0.85|7,146.6|14.77|0.0143|
> |qwen3-30b-a3b|standard|0.52|3,089.1|12.59|0.0013|
> |qwen3-30b-a3b|pes@12|0.77|40,666.5|126.95|0.0175|
> |qwen3-30b-a3b|prog@3/6|0.79|11,680.1|31.17|0.0049|
>
> In fact all methods run reviews in parallel with the same concurrency, and for LLMs the token usage can reflect the total runtime and API costs very well. Although the actual runtime and cost may vary depending on the hardware used in the experiments, both are largely proportional to our chosen efficiency metric, i.e., eq. output tokens. The conclusions from these additional experiments are consistent with Figure 4: at comparable performance, the progressive method is approximately 4× more efficient than the simple pessimistic verification approach commonly adopted in agentic workflows for math.
>
> As for the annotation-error claim, we have provided several examples in Appendix A.3, and a full list of our manual reviews in the supplimentary material. We only conducted a small number of manual reviews due to the limitation of expertise labor, but the error proportion already showed significant evidence of the benchmark unreliability. For example, around 80% of the FNs of Gemini 3 Pro Preview is caused by annotation-error.
>
> The annotation-errors we claim is easy to reproduce and objectively exist in these manually labeled datasets, so we welcome any individual adjudication study on them. It will be benificial for this research area as well.
>
> hopefully this can adress some of your concerns!
>
> [1] Hard2Verify. http://arxiv.org/abs/2510.13744
>
> [2] Open Proof Corpus. http://arxiv.org/abs/2506.21621
>
> [3] DeepSeekMath-V2. http://arxiv.org/abs/2511.22570
>
> [4] Gemini 2.5 Pro Capable of Winning Gold at IMO 2025. http://arxiv.org/abs/2507.15855
>
> [5] AI Mathematician. http://arxiv.org/abs/2505.22451

---

> > ### Author Rebuttal · Reviewer_BdY7 · 2026-04-04
> >
> > Thank you for your rebuttal. The rebuttal provides clarifications on several points, but does not fully resolve my main concerns.
> >
> > On evaluation choices, the authors clarify that progressive@3/6 and the 300-example subset were selected for cost reasons and confirm that no validation or tuning protocol was used. While this improves transparency, it does not address the potential for selection bias or the lack of a principled evaluation setup.
> >
> > The efficiency claim is partially strengthened by additional runtime and cost numbers, which are helpful. However, the analysis still lacks details on parallelization settings and variability, and the reliance on token-based proxies remains only partially justified.
> >
> > More importantly, several core concerns remain unresolved. The lack of comparisons to stronger verifier/grading baselines is not convincingly addressed; arguing that these methods have different objectives does not fully justify their exclusion, especially given that they are cited as closely related work. The annotation-error claim also remains insufficiently supported, as it still relies on a small, non-independent manual analysis without broader adjudication or corrected metrics. Finally, the concern about QiuZhen-Bench being labeled with the authors’ own method is not directly addressed.
> >
> > Overall, while the rebuttal improves clarity and provides some additional evidence, it does not substantially change my assessment of the paper’s empirical rigor or the strength of its claims.

---

> > > ### Author Response · Authors · 2026-04-06
> > >
> > > Thank you again for your time and effort spared in reviewing this work! Here we will provide more things that might further address some of your concerns.
> > >
> > > **Evaluation choices and efficiency claims**
> > >
> > > Our hyperparameter choices were primarily determined based on efficiency experiments conducted on GPT-5-mini and Qwen3-30B-A3B, and therefore may not be optimal for other models. However, we emphasize that no deliberate manipulation was involved, and any potential bias is minimal and unlikely to affect the main conclusions. In our efficiency experiments, we set the API call concurrency to 8. Since our method executes API calls in a fully parallel manner, increasing concurrency would approximately reduce runtime proportionally, while keeping both cost and token consumption unchanged. Token usage is indeed an effective metric for evaluating the efficiency of large language model applications.
> > >
> > > **Comparing with grading workflows**
> > >
> > > As noted earlier, the related work we cited does not address exactly the same task as ours. Their setting focuses on aligning with subjective grading preferences, whereas ours is concerned with distinguishing correct from incorrect proofs. That said, if a comparison is deemed necessary, the scores produced by those workflows can be interpreted as binary correctness judgments, from which the balanced F1 score can be computed. To this end, we used the codebase of a recent representative work, ProofGrader[1], and evaluated the implemented workflows on GPT-5-mini and the Hard2Verify dataset for comparison with our method. The results are as follows:
> > >
> > > |Method|bal. F1|TNR|eq. Token Usage / sample|
> > > |-|-|-|:-:|
> > > |single-pass|0.796|0.684|2895.18|
> > > |proofgrader vanilla|0.807|0.688|3362.27|
> > > |decompose-then-judge|0.742|0.601|7998.73|
> > > |reflect-and-revise|0.769|0.646|9078.97|
> > > |repeat-and-aggregate|0.799|0.677|9710.45|
> > > |prog@3/6|**0.872**|**0.842**|7088.45|
> > >
> > > In the comparison, we treated all non-perfect scores produced by these workflows as incorrect. Even under this favorable interpretation, their TNR and overall balanced F1 show little to no improvement and remain clearly inferior to our method. This is also consistent with observations made in prior dataset work: parallel workflows do not effectively improve verification capability. This does not mean that these related works are useless, instead they can increase the alignment of LLM grading with that of humans.
> > >
> > > Moreover, as discussed earlier in the rebuttal, the various verification-assisted solving methods we cited can all be viewed as variants of simple pessimistic verification, and have already been thoroughly compared in the solving experiments.
> > >
> > > **Insufficient support of annotation-error claims**
> > >
> > > Regarding potential annotation errors in the dataset, we manually verified only ~100 samples that were judged incorrect by large models. However, for GPT-5-mini and stronger models, the mislabeling rate among these cases already exceeds 50%. Based on this, we infer that these verification benchmarks have become substantially unreliable for stronger models. We therefore further evaluated frontier models such as Gemini 3 Pro on more challenging problem-solving datasets, including IMO and Apex. In the assisted problem-solving setting, our method achieves over 2× efficiency gains along with consistent performance improvements compared to existing approaches, whereas such advantages are not reflected on the verification benchmarks.
> > >
> > > We acknowledge that, within the rebuttal period, we are unable to conduct a larger-scale independent validation. Nevertheless, we have made our best effort to document the identified cases in the appendix and supplementary material. IMO Grading Bench is a widely recognized dataset released by Google, and pointing out annotation errors is not a claim we make lightly. We stand by our findings, but at this stage, this is the most comprehensive evidence we are able to provide.
> > >
> > > **Unreliability of QiuZhen-Bench**
> > >
> > > Please refer to section Q2 & W3 of the rebuttal to reviewer o8pG. Other reviewers also proposed the same concern while for them it is considered as resolved. I hope that could be convincing for you as well.
> > >
> > > **Conclusion**
> > >
> > > Thank you for taking the time to carefully read this rebuttal. We also acknowledge that pessimistic verification is not a perfect work, and it may not fully address all of your concerns. Nevertheless, we believe the proposed method is simple, effective, and readily applicable to many AI4Math tasks. We also hope that several of our findings, such as the error-centered perspective on verification, the stability of pessimistic verification, and the empirical observations on parallel scaling, may provide valuable insights for future research.
> > >
> > > [1] https://github.com/euclidgame/proofgrader

---

### Official Review · Reviewer_YM3F · 2026-03-13

**Soundness:** 4
**Presentation:** 4
**Significance:** 2
**Originality:** 2
**Overall Recommendation:** 4
**Confidence:** 4

**Summary:**

The paper studies LLM-based verification for open-ended mathematical proofs and argues that the main failure mode is not recognizing correct solutions, but reliably detecting subtle errors; motivated by this view, it introduces _pessimistic verification_, in which a proof is rejected if any verifier flags a flaw, along with a stronger _progressive_ variant that recursively decomposes the proof and verifies it at multiple granularities. Empirically, the authors report consistent gains over standard verification, majority-vote style aggregation, and longer chain-of-thought on Hard2Verify, IMO-GradingBench, and an additional advanced-math benchmark, with especially strong improvements in true-negative rate and overall efficiency; they further argue, via manual case analysis, that benchmark label noise obscures the method’s actual performance on stronger models. The paper also includes downstream solving experiments on IMO 2025 and MathArena Apex 2025, where progressive pessimistic verification appears to improve both accuracy and token efficiency in iterative self-correction workflows. Overall, the central message is that error-centric, fine-grained verification may be a more effective test-time scaling strategy for math reasoning than simply extending generation length.

**Compliance With Llm Reviewing Policy:**

Affirmed.

**Key Questions For Authors:**

Would the authors be willing to add experiments examining the relationship between the effectiveness of pessimistic verification and the quality of the underlying single-turn verifier? Such an analysis would help clarify the conditions under which the proposed approach is most effective.

**Strengths And Weaknesses:**

The empirical results appear sound and are largely consistent with intuition. In particular, the proposed approach seems practically useful for designing scaffolds or test-time frameworks for mathematical problem solving. That said, the level of novelty appears somewhat limited. The core idea behind this style of verification appears related to mechanisms already mentioned in prior work, for example in Zhang et al. (2025) [arXiv:2507.15855], so the main contribution may lie more in the empirical validation than in the underlying concept itself.

The paper is generally clear and well presented. One aspect that would strengthen it substantially is a more explicit analysis of the relationship between pessimistic verification and the verifier’s underlying single-turn performance. In particular, the effectiveness of a pessimistic strategy would seem to depend strongly on the base verifier’s error profile: for example, if a model has a high false-negative rate, one might expect pessimistic verification to be less effective or even counterproductive. A systematic investigation of this dependency would help clarify when the method should be expected to work well and would significantly improve the paper.

---

> ### Author Rebuttal · Authors · 2026-03-28
>
> Thank you for your approval for our work! Here are some further explanations of some details in this work.
>
> Firstly, let's discuss the novelty of this work over the existing methods. It is intuitive to use multiple parallel reviews on a given solution, and try to resolve all issues discovered by these reviewers to improve the output quality. So many math problem-solving workflows already adopts similar methods to construct their reviewer, which can be viewed as a variant of simple pessimistic verifier. We also cited some examples in our paper [1][2][3]. The problem is, **these methods are not viewed as a verification method**. i.e., their primary purpose is to use them to assist math problem-solving, and discover as many potential issues as possible, but the ability on preserving true positive rate is not considered.
>
> Similar perspectives also exist in related benchmark papers, such as Hard2Verify[4] and OPC[5], and in Hard2Verify the authors even claimed that "We find sequential scaling (long-CoT) brings substantive gains, whereas parallel scaling does not.". They primarily evaluated two types of parallel scaling, majority voting and best-of-N selection after running $N$ parallel verification, and obtained no performance gains. However, methods like simple pessimistic verification were not considered. Intuitively this method might be effective to increase the TNR of verification, but it is likely that the TPR would severely drop, and this will not bring any increment of the overall balanced F1 score.
>
> So one of our contributions is that, we have confirmed a counterintuitive result, that in most cases the TPR is actually very stable under pessimistic verification. As we increase the number of parallel verifiers we could obtain steady gains of balanced F1 score, making it an effective "verification method". Its effective even surpassed long-CoT mentioned in Hard2Verify. There is only a small drop of TPR with high parallel settings, and a large amount of that might actually be caused by annotation noise.
>
> We also introduced a new progressive verification method, with the intuition of how we are doing reviews on code or math proofs, where a fine-grained review into the details will be much more effective than doing overall reviews for multiple times. It has exhibited significantly higher performance and efficiency over previous simple methods, across many verification benchmarks and problem-solving benchmarks.
>
> Then here comes another question: under which condition will pessimistic verification methods be effective, or what is the relationship between its effectiveness and the underlying LLMs?
>
> In fact in *Figure 5* we have already provided a comprehensive analysis of the verification performance of many different LLMs, from GPT-5 to Qwen3-1.7B, with standard single-pass verification and progressive@3/6 setting. The base verifier's error profile can be easily infered from the standard TPR and standard TNR provided in that graph. We have listed some results in page-5 in our paper, and we can still learn more from these data:
>
> * **Pessimistic verification is almost always effective** across all models in our experiments, as the balanced F1 of them almost always increases.
> * TNR is the key ability that separates strong verifiers from the weak, which can be significantly increased by pessimistic verification.
> * TPR is almost the same across all models from strong to weak.
> * TPR is stable for most models under progressive@3/6, and small drops in TPR under strong underlying models can mostly be attributed to annotation errors in the dataset.
> * However, in weaker models like Qwen3-8B, Qwen3-4B, and Qwen3-1.7B, we also witnessed notable decrement of TPR under pessimistic verification. This decrement is more severe for weaker models.
>
> So this indicates that, pessimistic verification might still be counterproductive for smaller models under extremely high settings. The upper bound of parallel verifications the method could bear is still limited by the consistency on positive examples of underlying model. However in our continued efficiency experiments, for models like Qwen3-30B-A3B, we also obtained positive performance gains under progressive@5/6, which applies up to 31 parallel verifications. This upper bound would be much higher for frontier LLMs like Gemini 3 Pro or GPT-5.4. Cost would be the primary limitation in application. And due to the excessive cost and unreliability of the benchmarks, we have not tested their actual performance upper bound under pessimistic verification methods.
>
> I hope this could resolve your concerns about this work!
>
> [1] DeepSeekMath-V2. http://arxiv.org/abs/2511.22570
>
> [2] Gemini 2.5 Pro Capable of Winning Gold at IMO 2025. http://arxiv.org/abs/2507.15855
>
> [3] AI Mathematician. http://arxiv.org/abs/2505.22451
>
> [4] Hard2Verify. http://arxiv.org/abs/2510.13744
>
> [5] Open Proof Corpus. http://arxiv.org/abs/2506.21621

---

> > ### Author Rebuttal · Reviewer_YM3F · 2026-04-06
> >
> > Thank you for the detailed clarification and for elaborating on the distinctions you see in the work. I understand your point more clearly now, especially regarding the framing of pessimistic verification as a verification method and your empirical findings on TPR stability and balanced F1 improvements.
> >
> > That said, my main concern remains that this general verification pattern and its practical effectiveness already seem to be fairly well understood in the community, at least as a component within broader workflows. Since methods of this flavor are already being used in related settings, I am not fully convinced that the core idea itself is sufficiently novel. For that reason, I would keep my current score unchanged.

---

> > > ### Author Response · Authors · 2026-04-06
> > >
> > > Dear Reviewer,
> > > Thank you for your thoughtful comments. We understand your concerns about this work. While we previously spent substantial space introducing the background, we would like to further clarify the novelty and contributions of our paper.
> > >
> > > * Direct multi-round verification for mathematical problems is an intuitive strategy. Although related works have used similar ideas, their stability and reliability have not been systematically studied. As a result, such methods have generally been treated as auxiliary heuristics rather than robust verification approaches, and their actual effectiveness for math verification remained unclear. Clarifying this point is one of our key contributions.
> > > * At the same time, the methods used in prior work can largely be viewed as variants of simple pessimistic verification. Taking this as the baseline, our proposed multi-scale progressive pessimistic verification still achieves 2–4× higher efficiency, along with consistent performance gains. This can be seen clearly in Figures 4 and 7. We believe this constitutes another important contribution of our work.
> > > * To the best of our knowledge, the design of progressive pessimistic verification has not been explored before. Moreover, the broad use of simple verification strategies in mathematical reasoning workflows further highlights the significance of our study. Both our analysis of the simple variant itself and the proposed progressive extension can be directly applied to future AI4Math research, or at least provide useful inspiration for related work.
> > >
> > > Thank you again for taking the time to read our response. We hope this clarifies both the relationship between our work and prior research, as well as the specific contributions we make. We believe this work is valuable for the community and worthy of further attention and application, and we would sincerely appreciate your thoughts.

---

### Decision · Program_Chairs · 2026-04-30

**Decision:**

Accept (regular)

**Comment:**

Reviewers find the idea useful but largely incremental. Concerns about evaluation (limited baselines, unclear design choices, and weak support for key claims) remain after rebuttal. Overall, the evidence is not strong enough for acceptance.